# Heads collapse, features stay: Why Replay needs big buffers

**Giulia Lanzillotta**,* **Damiano Meier**\* **& Thomas Hofmann**
ETH AI Center & Department of Computer Science, ETH Zürich

## Abstract

A persistent paradox in continual learning (CL) is that neural networks often retain linearly separable representations of past tasks even when their output predictions fail. We formalize this distinction as the gap between *deep* (feature-space) and *shallow* (classifier-level) forgetting. We reveal a critical asymmetry in Experience Replay: while minimal buffers successfully anchor feature geometry and prevent deep forgetting, mitigating shallow forgetting typically requires substantially larger buffer capacities. To explain this, we extend the Neural Collapse framework to the sequential setting. We characterize deep forgetting as a geometric drift toward out-of-distribution subspaces and prove that any non-zero replay fraction asymptotically guarantees the retention of linear separability. Conversely, we identify that the "strong collapse" induced by small buffers leads to rank-deficient covariances and inflated class means, effectively blinding the classifier to true population boundaries. By unifying CL with out-of-distribution detection, our work challenges the prevailing reliance on large buffers, suggesting that explicitly correcting these statistical artifacts could unlock robust performance with minimal replay.

## 1 Introduction

Continual learning (Hadsell et al., 2020) aims to train neural networks on a sequence of tasks without catastrophic forgetting. It holds particular promise for adaptive AI systems, such as autonomous agents that must integrate new information without full retraining or centralized data access. The theoretical understanding of optimization in non-stationary environments remains limited, particularly regarding the mechanisms that govern the retention and loss of learned representations.

A persistent observation in the literature is that neural networks retain substantially more information about past tasks in their internal representations than in their output predictions. This phenomenon, first demonstrated through *linear probe evaluations*, shows that a linear classifier trained on frozen last-layer representations achieves markedly higher accuracy on old tasks than the network's own output layer (Murata et al., 2020; Hess et al., 2023). In other words, past-task data remain linearly separable in feature space, even when the classifier fails to exploit this structure. This motivates a distinction between two levels of forgetting: **shallow forgetting**, corresponding to output-level degradation recoverable by a linear probe, and **deep forgetting**, corresponding to irreversible loss of feature-space separability.

In this work, we show that *replay buffers affect these two forms of forgetting in systematically different ways*. Replay—the practice of storing a small subset of past samples for joint training with new data—is among the most effective and widely adopted strategies in continual learning. However, the requirement to store and repeatedly process substantial amounts of past data limits its scalability. Our analysis reveals a critical efficiency gap: *even small buffers are sufficient to preserve feature separability and prevent deep forgetting*, whereas mitigating shallow forgetting requires substantially larger buffers. Thus, while replay robustly preserves representational geometry, it often fails to maintain alignment between the learned head and the true data distribution.

To explain this phenomenon, we turn to the geometry of deep network representations. Recent work has shown that, at convergence, standard architectures often exhibit highly structured, low-

---

*Equal contribution, email at {glanzillo,dammeier}@ethz.ch.

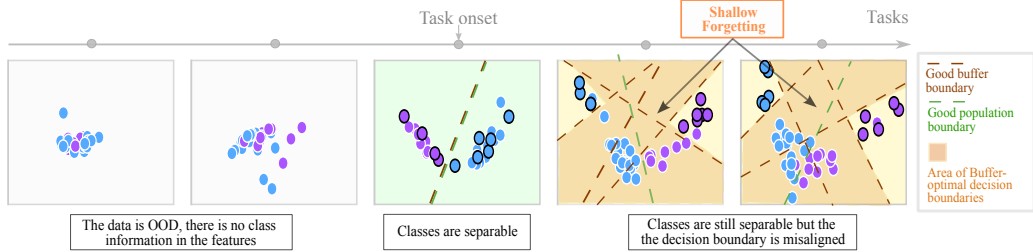

Figure 1: **Evolution of decision boundaries and feature separability.** PCA evolution of two Cifar10 classes (1% replay). Replay samples are highlighted with a black edge. While features retain separability across tasks (low deep forgetting), the classifier optimization becomes *under-determined*: multiple "buffer-optimal" boundaries (dashed brown) perfectly classify the stored samples but largely fail to align to the true population boundary (dashed green), resulting in shallow forgetting.

dimensional feature organization. In particular, the *Neural Collapse* (NC) phenomenon (Papyan et al., 2020) describes a regime in which within-class variability vanishes, class means form an equiangular tight frame (ETF), and classifier weights align with these means. Originally observed in simplified settings, NC has now been documented across architectures, training regimes, and even large-scale language models (Súkeník et al., 2025; Wu & Papyan, 2025), making it a powerful framework to analyze feature-head interactions.

In this work, we extend the NC framework to continual learning, providing a characterization of the geometry of features and heads under extended training. Our analysis covers task-, class-, and domain-incremental settings and explicitly accounts for replay. To account for this, we propose *two governing hypotheses* for historical data absent from the buffer: (1) topologically, forgotten samples behave as out-of-distribution (OOD) entities, and (2) enlarging the replay buffer induces a smooth interpolation between this OOD regime and fully collapsed representations. These insights allow us to construct a simple yet predictive theory of feature-space forgetting that lower-bounds separability and captures the influence of weight decay, feature-norm scaling, and buffer size.

In summary, this paper makes the following distinct contributions:

1. **The replay efficiency gap.** We identify an *intrinsic asymmetry* in replay-based continual learning: minimal buffers are sufficient to anchor feature geometry (preventing deep forgetting), whereas mitigating classifier misalignment (shallow forgetting) requires disproportionately large capacities.

2. **Asymptotic framework for continual learning.** We extend Neural Collapse theory to continual learning, characterizing the asymptotic geometry of both single-head and multi-head architectures and identifying unique phenomena like rank reduction in task-incremental learning.

3. **Effects of replay on feature geometry.** We demonstrate that shallow forgetting arises because classifier optimization on buffers is *under-determined*—a condition structurally *exacerbated by Neural Collapse*. The resulting geometric simplification (covariance deficiency and norm inflation) blinds the classifier to the true population boundaries.

4. **Connection to OOD detection.** We re-conceptualize deep forgetting as a geometric drift toward out-of-distribution subspaces. This perspective bridges the gap between CL and OOD literature, offering a rigorous geometric definition of "forgetting" beyond simple accuracy loss.

## 1.1 NOTATION AND SETUP

We adopt the standard compositional formulation of a neural network, decomposing it into a feature map and a classification head. The network function is defined as $f_\theta(x) = h(\phi(x))$, where $h(z) = W_h z + b_h$, with parameters $\theta = \{\phi, W_h\}$.

We refer to $\phi$ as the *feature map*, to its image as the *feature space* and to $\phi(x)$ as the *features* or *representation* of input $x$.

We consider sequential classification problems subdivided into tasks. For each class $c$, a dataset of labeled examples $(X_c, Y_c)$ is available for training. Given any sample $(x, y)$, the network prediction is obtained via the maximum-logit rule

$$\hat{y} = \arg\max_k \langle w_k, \phi(x) \rangle,$$

where $w_k$ denotes the $k$-th column vector of $W_h$. Network performance is evaluated after each task on all previously seen tasks.

Following Lopez-Paz & Ranzato (2017), *shallow forgetting* is quantified as the difference $A_{ij} - A_{jj}$, where $A_{ij}$ denotes the accuracy on task $j$ measured after completing learning session $i$. In contrast, *deep forgetting* is defined as the difference $A_{ij}^\star - A_{jj}^\star$, where $A_{ij}^\star$ represents the accuracy of a *linear probe* trained on the frozen representations of task $j$ at the end of session $i$.

We adopt the three continual learning setups introduced by van de Ven et al. (2022), described in detail in Section 3: *task-incremental learning* (TIL), *class-incremental learning* (CIL), and *domain-incremental learning* (DIL).

For the experimental analysis, we train both ResNet and ViT architectures, from scratch and from a pre-trained initialization. We train on three widely used benchmarks adapted to the continual learning setting: *Cifar100* (Krizhevsky & Hinton, 2009), *Tiny-ImageNet* (Torralba et al., 2008), and *CUB200* (Wah et al., 2011). A detailed description of datasets and training protocols, including linear probing, is provided in Appendix A.1.

## 2 EMPIRICAL CHARACTERIZATION OF DEEP AND SHALLOW FORGETTING

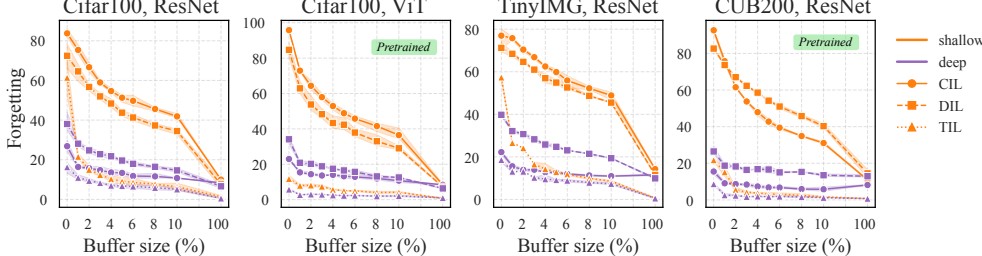

Figure 2: **Replay efficiency gap.** Forgetting decays at different rates in the feature space and the classifier head, producing a persistent gap between deep and shallow forgetting. Increasing the replay buffer closes this gap only gradually, with substantial buffer sizes required for convergence. See Appendix A.2 for details.

We first present our main empirical finding. We evaluate forgetting in both the network output layer and a linear probe trained on frozen features across varying buffer sizes, datasets, and architectures (randomly initialized and pre-trained). Our results, summarized in Figure 2, reveal a robust phenomenon: **while small replay buffers are sufficient to prevent *deep forgetting* (preserving feature separability), mitigating *shallow forgetting* requires substantially larger buffers**. This extends prior observations of feature-output discrepancies (Murata et al., 2020; Hess et al., 2023) by demonstrating that *replay stabilizes representations far more efficiently than it maintains classifier alignment*. The gap persists across settings, vanishing only near full replay (100%).

We highlight three specific trends:

1. *Head architecture.* The deep–shallow gap is pronounced in single-head setups (CIL, DIL) but significantly smaller in multi-head setups (TIL).

2. *Replay efficacy in DIL.* Contrary to the assumption that CIL is the most challenging benchmark, DIL exhibits high levels of deep forgetting, converging to levels similar to CIL.

3. *Pre-training robustness.* Corroborating Ramasesh et al. (2021), pre-trained models exhibit negligible deep forgetting. Their feature spaces remain robust even with minimal replay, yielding nearly flat deep-forgetting curves.

In the following section, we present a theoretical model explaining this *asymmetric effect* of replay via the asymptotic dynamics of the feature space.

## 3 NEURAL COLLAPSE UNDER SEQUENTIAL TRAINING

### 3.1 PRELIMINARIES ON NEURAL COLLAPSE

Recent work (Papyan et al., 2020; Lu & Steinerberger, 2022) characterizes the geometry of representations in the *terminal phase of training* (TPT) the regime in which the training loss has reached zero and features stabilize. In this regime, features converge to a highly symmetric configuration known as *Neural Collapse* (NC), which is provably optimal for standard supervised objectives and emerges naturally under a range of optimization dynamics (Tirer & Bruna, 2022; Súkeník et al., 2025).

We denote the feature class means by $\mu_c(t) = \mathbb{E}_{x \in X_c}[\phi_t(x)]$, $\tilde{\mu}_c(t)$ the centered means, and the matrix of centered means by $\tilde{U}(t)$. We focus on first three properties defining NC:

- $\mathcal{NC}1$ **(Variability Collapse).** Within-class variability vanishes as features collapse to their class means: $\phi_t(x) \to \mu_c(t), \forall x \in X_c$, implying the within-class covariance approaches **0**.

- $\mathcal{NC}2$ **(Simplex ETF).** Centered class means form a simplex Equiangular Tight Frame (ETF). They attain equal norms and maximal pairwise separation:

$$\lim_{t \to \infty} \langle \tilde{\mu}_c(t), \tilde{\mu}_{c'}(t) \rangle = \begin{cases} \beta_t & \text{if } c = c' \\ -\frac{\beta_t}{K-1} & \text{if } c \neq c' \end{cases}$$

- $\mathcal{NC}3$ **(Neural Duality).** Classifier weights align with the class means up to scaling, i.e., $W_h^\top(t) \propto \tilde{U}(t)$.

### 3.2 NEURAL COLLAPSE IN CONTINUAL LEARNING

Standard evaluation in continual learning measures performance strictly at the completion of each task. Thus, while forgetting arises from optimization dynamics, its magnitude is defined effectively by the network's configuration at the end of training. We leverage the Neural Collapse framework to rigorously characterize this terminal geometry, modeling the stable structures that emerge in the limit of long training times.[1]

While prior work focuses on stationary settings, we extend the NC framework to continual learning. We empirically verify its emergence in domain- (DIL), class- (CIL), and task-incremental (TIL) settings (Figure 3, see Appendix C.4).

*Observed vs. population statistics.* NC emerges on the training data (current task + buffer). We must therefore distinguish between *observed* statistics $\hat{\mu}$ (computed on available training samples) and *population* statistics $\mu$ (computed on the full distribution). The following empirical analysis concerns $\hat{\mu}$; in subsequent sections, we develop a theory for $\mu$ to quantify forgetting.

#### 3.2.1 SINGLE-HEAD ARCHITECTURES

In **domain-incremental learning (DIL)**, all tasks share a fixed label set, with each task introducing a new input distribution. Consequently, while the estimated class means $\hat{\mu}_c$ and global mean $\hat{\mu}_G$ evolve throughout the task sequence, the asymptotic target geometry remains invariant: the number of class means and their optimal angles are constant. We find that the NC properties established in

---

[1]Our analysis focuses on structures at convergence; however, we observe that Neural Collapse emerges quickly in practice across standard architectures (cf. Figure 3).

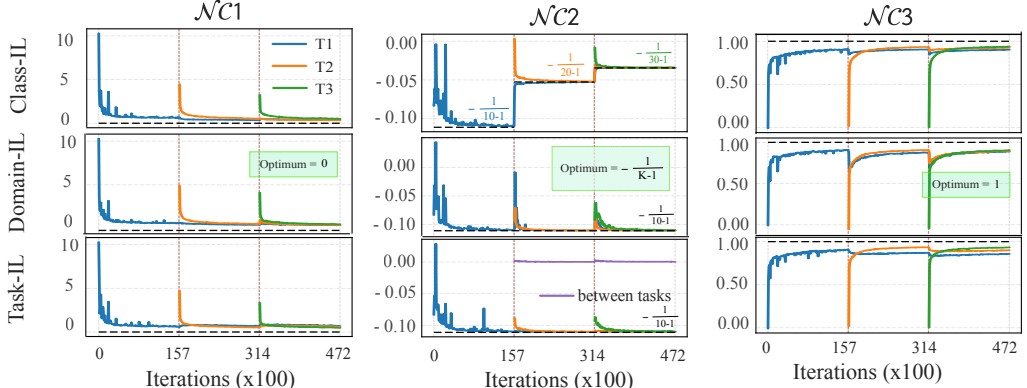

Figure 3: **NC metrics in sequential training (Cifar100, ResNet with 5% replay).** NC emerges across all tasks. In DIL, the ETF structure ($\mathcal{NC}2$) remains stable; in CIL, it evolves as class count increases; in TIL, it arises per-head with variable cross-task alignment. Highlighted in green is the asymptotic limit of the NC metrics. See Appendix A.2 for details.

the single-task regime (Definitions 4 to 6) persist under DIL. When a replay buffer is employed, the class means are effectively computed over the mixture of new data and buffered samples.

In **class-incremental learning (CIL)**, each task introduces a disjoint subset of classes. The asymptotic structure of the feature space is therefore redefined after each task, governed by the relative representation of old versus new classes. When past classes are under-represented in the training dataset, they act as *minority classes*: their features collapse toward a degenerate distribution centered near the origin, and their classifier weights converge to constant vectors (Fang et al., 2021; Dang et al., 2023). This phenomenon, known as *Minority Collapse* (MC), occurs sharply below a critical representation threshold. Without replay, MC dominates the asymptotic structure as past classes are absent from the loss. However, we observe that replay mitigates this effect when buffers are sampled in a *task-balanced manner*. This strategy ensures that all classes—both new and old—are equally represented in each training batch, thereby preserving the global ETF structure and preventing the marginalization of past tasks (Figure 3).

### 3.2.2 MULTI-HEAD ARCHITECTURES

Neural Collapse has not previously been characterized in *multi-head* architectures. In **task-incremental learning (TIL)**, the network output is partitioned into separate heads, each associated with a distinct task. This ensures that error propagation is localized to the assigned head (see Figure 25). While this local normalization prevents Minority Collapse even without replay, the resulting global geometry across tasks is non-trivial. Specifically, we investigate the relative angles and norms between class means belonging to different tasks.

We measure standard NC metrics including within-class variance, inter-task inner products, and feature norms. Our findings reveal a clear distinction between local and global structure in TIL:

1. *Local collapse.* NC emerges consistently *within* each head. Each task-specific head satisfies $\mathcal{NC}1$–$\mathcal{NC}3$ locally.

2. *Global misalignment.* A coherent cross-task NC structure is absent. Across tasks, class means display variable scaling and alignment (Figure 3, Figures 12 to 14 ).

3. *Rank reduction.* We find that local normalization induces a dimensionality reduction in the feature space. The global feature space attains a maximal rank of $n(K-1)$ for $n$ tasks, which is strictly lower than the $nK-1$ rank observed in single-head settings (Figure 16).

These empirical observations—specifically that task-balanced replay restores global NC in single-head setups while TIL lacks global alignment—serve as the foundation for the theoretical model of class separability developed in the next section.

## 4 ASYMPTOTIC BEHAVIOUR OF DEEP AND SHALLOW FORGETTING

### 4.1 PRELIMINARIES

**Linear separability.** To analyse deep forgetting, we require a mathematically tractable measure of linear separability in feature space. Formally, linear separability between two distributions $P_1$ and $P_2$ is the maximum classification accuracy achievable by any linear classifier. Given the first two moments $(\mu_1, \Sigma_1)$ and $(\mu_2, \Sigma_2)$, the Mahalanobis distance is a standard proxy. Here, we use the *signal-to-noise ratio* (SNR) between class distributions, defined as

$$SNR(c_1, c_2) = \frac{\|\mu_1 - \mu_2\|^2}{\text{Tr}(\Sigma_1 + \Sigma_2)}.$$

Higher SNR values imply greater separability. In Appendix C.2, we show that this quantity lower-bounds the Mahalanobis distance, and thus linear separability itself. Accordingly, we focus on the first- and second-order statistics of class representations (means and covariances), as these directly govern the SNR.

**Asymptotic notation.** We use $\mathcal{O}(\cdot)$ and $\Theta(\cdot)$ to characterize the scaling of time-dependent quantities $f(t)$, suppressing constants independent of $t$. When bounds depend on controllable quantities such as the buffer size $b$, we retain these dependencies explicitly. This notation highlights scaling behaviour relevant to training dynamics and experimental design choices.

### 4.2 ANALYSIS OF DEEP FORGETTING

#### 4.2.1 FORGOTTEN ≈ OUT-OF-DISTRIBUTION

The Neural Collapse (NC) framework characterizes the asymptotic geometry of representations for *training* data. Forgetting, however, concerns the evolution of representations for samples of past tasks that are no longer part of the optimization objective. We bridge this conceptual gap through the following hypothesis:

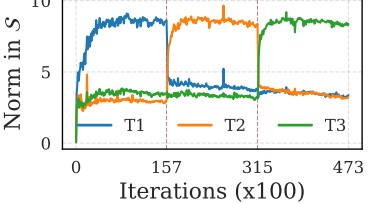

**Hypothesis 1.** Forgotten samples behave analogously to samples that were never learned, i.e., they are effectively *out-of-distribution* (OOD) with respect to the current model.

Figure 4: **Projection of $\tilde{\mu}_c(t)$ onto $S_t$ (Cifar100, no replay)**. The population means of past and future tasks exhibit equivalent (near-zero) norms when projected onto the active subspace $S_t$.

This perspective motivates our analysis of forgetting as a form of *shift to out-of-distribution* in feature space. Specifically, in the absence of replay, data from past tasks exhibits the same geometric behaviour as future-task (OOD) inputs. To formalize this correspondence, we adopt a feature-space definition of OOD based on the recently proposed *ID/OOD orthogonality* property (*NC5*, Ammar et al., 2024).

**Definition 1** (Out-of-distribution (OOD)). Let $X_c$ denote the samples of class $c$, and let $\phi_t(x)$ be the feature map of a network trained on dataset $D$ with $K$ classes. Denote by $S_t = \text{span}\{\tilde{\mu}_1(t), \ldots, \tilde{\mu}_K(t)\}$ the **active subspace** spanned by the *centered* class means of the training data at time $t$. We say that $X_c$ is *out-of-distribution* for $\phi_t$ if the average representation of $X_c$ is orthogonal to $S_t$.

In Appendix C.6 (Proposition 2), we show that, under the NC regime, the empirical observation that OOD inputs yield higher predictive entropy than in-distribution (ID) inputs is mathematically equivalent to this orthogonality condition—thus establishing a formal connection between predictive uncertainty and the geometric structure of NC5.

We validate our hypothesis by monitoring the projection of centered class means $\tilde{\mu}_c$ onto the active subspace $S_t$. As shown in Figure 4 (and Figures 17 to 19), shortly after a task switch, the projection of past-task means *collapses sharply*, indistinguishably matching the behavior of unseen (OOD) tasks.

### 4.2.2 ASYMPTOTIC DISTRIBUTION OF OOD CLASSES

Leveraging the connection between forgetting and OOD dynamics, we now characterize the asymptotic behavior of past-task data. We find that *the residual signal of past classes is confined to the inactive subspace $S^\perp$*, making it susceptible to erasure by weight decay.

**Theorem 1** (Asymptotic distribution of OOD data). *Let $X_c$ be OOD inputs (Definition 1) for a feature map $\phi_t$ trained with a sufficiently small learning rate $\eta$ and weight decay $\lambda$. Let $\beta_t$ denote the observed centered class-mean norm as by Definition 5. In the terminal phase ($t \geq t_0$), the feature distribution of $X_c$ has mean $\mu_c$ and variance $\sigma_c^2$ given by:*

$$\mu_c(t) = (1 - \eta\lambda)^{t-t_0} \, \mu_{c,S^\perp}(t_0), \tag{1}$$

$$\sigma_c^2(t) \in \Theta\left(\beta_t + (1 - \eta\lambda)^{2(t-t_0)}\right). \tag{2}$$

**Corollary 1** (Collapse to null distribution). *If $\lambda > 0$, the OOD distribution converges to a degenerate null distribution: the mean decays to zero, and the variance limits depend on $\beta_t$.*

The proof (see Theorem 4) relies on the observation that, once $\mathcal{NC}3$ (alignment between class feature means and classifier weights) emerges, *optimization updates become restricted to the active subspace $S_t$*. Consequently, components of the representation in the orthogonal complement $S_t^\perp$ are frozen—or decay exponentially under weight decay— yielding the dynamics above.

☞ **Notation.** For brevity, let $\upsilon = 1 - \eta\lambda$, and note that $S_t = S_{t_0} = S$ for all $t \geq t_0$.

**Theorem 2** (Lower bound on OOD linear separability). *For two OOD classes $c, c'$ in the TPT, let $\upsilon = 1 - \eta\lambda$. The Signal-to-Noise Ratio (SNR), which lower-bounds linear separability, satisfies:*

$$\mathrm{SNR}(c, c') \in \Theta\left(\left(\frac{\beta_t}{\upsilon^{2(t-t_0)}} + 1\right)^{-1}\right).$$

**Discussion.** Theorem 2 does not imply that separability necessarily vanishes; consistent with our empirical findings (Figure 2), a residual signal persists in $S^\perp$. However, this signal is fragile. The result reveals the *dual role of weight decay*: it accelerates the exponential decay of the signal in $S^\perp$ (reducing the numerator), yet simultaneously prevents the explosion of the class-mean norm $\beta_t$ (constraining the denominator). Thus, weight decay both erases and indirectly preserves past-task representations.

We empirically observe that $\beta_t$ tends to increase upon introducing new classes (Appendix A.3.3), which amplifies forgetting, as by Theorem 2. We hypothesize this is an artifact of classifier head initialization in sequential settings. Preliminary experiments, discussed in Appendix A.3.3, lend support to this hypothesis; however, we leave a comprehensive investigation of this finding to future research.

### 4.2.3 ASYMPTOTIC DISTRIBUTION OF PAST DATA WITH REPLAY

Having seen that, without replay, past-task data behaves like OOD inputs drifting into $S^\perp$, we now consider how replay alters this picture. Replay provides a foothold in the active subspace $S$, preventing the collapse of old-task representations and preserving linear separability. Intuitively, the effect of replay should interpolate between the two extremes: no replay ($\mathcal{D}_{\mathrm{OOD}}$) and full replay ($\mathcal{D}_{\mathrm{NC}}$).

**Hypothesis 2.** The class structure in feature space emerges smoothly as a function of the buffer size, with past-task features retaining a progressively larger component in $S$.

To formalize this intuition, we introduce a mixture model for the asymptotic feature distribution under replay. Let $\pi_c \in [0, 1]$ denote a monotonic function of the buffer size, representing the fraction of the NC-like component retained in $S$. Then, in the terminal phase of training, the feature distribution of class $c$ can be expressed as a mixture

$$\phi(x) \sim \pi_c \, \mathcal{D}_{\mathrm{NC}} + (1 - \pi_c) \, \mathcal{D}_{\mathrm{OOD}}.$$

This model is exact in the extremes ($\pi_c = 0$ or $1$) and interpolates for intermediate buffer sizes.

**Validation.** Figure 5 confirms that increasing replay transfers variance from $S^\perp$ to $S$, improving separability. We observe that stronger weight decay reduces norms and within-class variability globally. Notably, we find an inverse relationship between buffer size and centered feature norms: while population means gravitate toward the global mean, representations in small-buffers are subject to a distinct *repulsive force*, pushing partially collapsed features outward. Finally, centered feature norms are consistently lower in DIL than in TIL or CIL.

This mixture model yields a lower bound on the Signal-to-Noise Ratio (SNR), proving that replay *guarantees* separability asymptotically.

**Theorem 3** (Lower bound on separability with replay). *Let $c, c'$ be past-task classes and $\pi \in (0, 1]$ the buffer mixing coefficient. In the TPT,*

$$\mathrm{SNR}(c, c') \in \Theta\left( \frac{r^2\,\beta_t + v^{2(t-t_0)}}{r^2\,\delta_t + \beta_t + v^{2(t-t_0)}} \right), \quad \text{where } r^2 = \frac{\pi^2}{(1 - \pi)^2}.$$

**Corollary 2.** *If $\pi > 0$ (non-empty buffer), the SNR does not vanish:* $\mathrm{SNR}(c, c') \in \Theta(r^2)$ *as $t \to \infty$.*

The corollary formalizes the intuition that any non-empty buffer anchors features in $S$. The anchoring strength $r^2$ grows with buffer size; empirically, this growth is superlinear in single-head models (CIL, DIL) but sublinear in multi-head TIL.

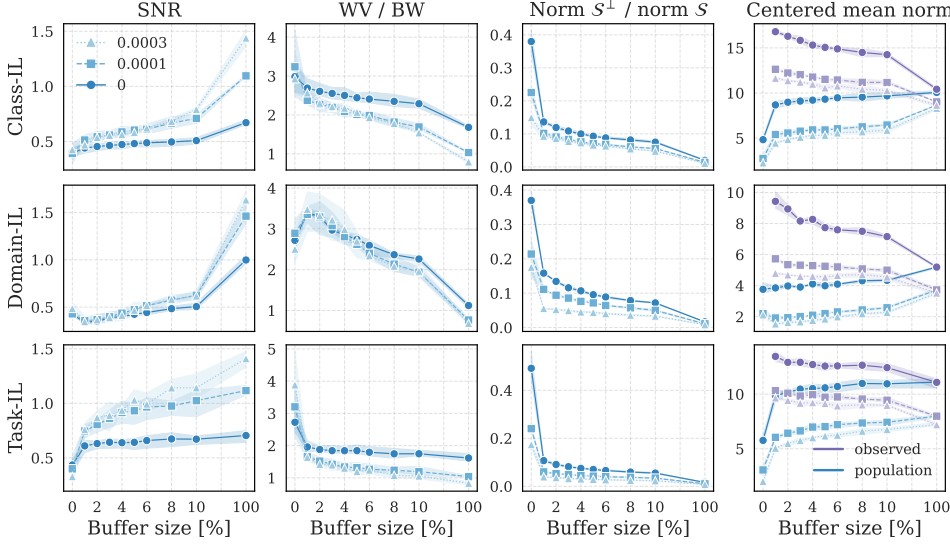

Figure 5: **Empirical validation for the theoretical model of feature space structure (Cifar100, ResNet with 5% replay).** Plot shows the average over all past tasks after training the last task for four metrics. Results are shown for different buffer sizes and weight decay parameters (different lines). Details in Appendix A.2.

**Discussion.** These results rigorously establish replay as an *anchor* within the active subspace $S$. While the absence of replay forces representations into $S^\perp$—causing exponential signal decay—any non-empty buffer guarantees a persistent signal proportional to $r^2$, ensuring asymptotic separability. Crucially, the efficiency of this anchoring varies by architecture: empirical trends (SNR, Figure 5) indicate *sublinear* growth of $\pi_c$ in single-head settings (CIL, DIL) versus *superlinear* growth in multi-head TIL, suggesting fundamental differences in how shared versus partitioned heads utilize replay capacity.

## 4.3 THE REPLAY EFFICIENCY GAP

We have established that even modest replay buffers suffice to anchor the feature space, preserving a non-vanishing Signal-to-Noise Ratio (mitigating *deep forgetting*). This resolves the first half of the puzzle. We now address the second half: *why does this preserved separability not translate into classifier performance (shallow forgetting)?*

**Mechanism: The under-determined classifier.** Shallow forgetting arises from the fundamental statistical divergence between the finite replay buffer and the true population distribution. This divergence is structurally amplified by Neural Collapse. As noted by Hui et al. (2022), small sample sizes induce a "strong" NC regime where samples collapse aggressively to their empirical means (yielding smaller $\mathcal{NC}1$ values, see Figure 15). Geometrically, this projects the buffer data onto a low-dimensional subspace $S_B \subset S$ (rank $\approx K - 1$). However, the true population retains variance in directions orthogonal to $S_B$ (specifically within $S^\perp$).

This geometric mismatch renders the optimization of the classifier head an **ill-posed, under-determined problem**. Let $W$ be the classifier weights. Since the buffer variance vanishes in directions orthogonal to $S_B$, the cost function is invariant to changes in $W$ along these directions. Consequently, the optimization landscape contains a manifold of "buffer-optimal" solutions that achieve near-zero training error. However, these solutions can vary arbitrarily in the orthogonal complement of $S_B$, leading to decision boundaries that are misaligned with the true population mass (as visualized in Figure 1). The classifier overfits the simplified geometry of the buffer, failing to generalize to the richer geometry of the population.

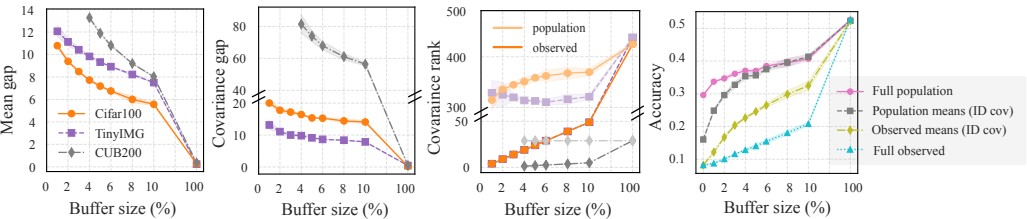

Figure 6: **Deconstructing the statistical gap.** *Left and center-left:* Gap (measured as L2 distance) between population and observed metrics. *Center-right:* Rank of the population (light shade) and observed (dark shade) covariance, the gap persists as the buffer size is increased. *Right:* Synthetic Linear Discriminant Analysis (LDA) on TinyIMG. We replace true statistics ($\mu, \Sigma$) with buffer estimates ($\hat{\mu}, \hat{\Sigma}$) to isolate error sources. Details in Appendix A.2.

**Mechanistic analysis of statistical divergence.** We quantitatively decompose this divergence into two primary artifacts, validated via synthetic Linear Discriminant Analysis (LDA) counterfactuals (Figure 6, methodological details in Appendix A.2). First, *covariance deficiency*: the buffer's empirical covariance $\hat{\Sigma}_B$ is rank-deficient and blind to variance in $S^\perp$. The criticality of second-order statistics is evidenced by the sharp accuracy drop observed when replacing the true population covariance with the identity matrix in LDA (gray line). Second, *mean norm inflation*: buffer means exhibit inflated norms relative to population means due to repulsive forces. Our LDA analysis confirms that replacing population means with buffer estimates (olive line) causes a distinct, additive performance degradation. Notably, when relying on both observed estimates of mean and covariance (cyan line) the performance of the LDA classifier drops below the original network's performance. Metrics such as mean and covariance gap (Figure 6, Left) further confirm that these discrepancies—particularly the covariance rank deficiency—persist until the buffer approaches full size.

**Implications.** These findings mechanistically explain the replay efficiency gap: the feature space retains linear separability, yet the classifier remains statistically blinded to it. Consequently, simply increasing buffer size is an inefficient, brute-force solution. Instead, our results suggest that *to bridge the gap between shallow and deep forgetting*, one must explicitly counteract the effects of Neural Collapse—specifically by preventing the extreme concentration and radial repulsion appearing in small buffers. We further elaborate on these implications in the discussion of future work.

## 5 RELATED WORK

Our work intersects three main research directions: the geometry of neural feature spaces, out-of-distribution (OOD) detection, and continual learning (CL). A more detailed overview is provided in Appendix B. Below we highlight the most relevant connections and our contributions.

**Deep vs. shallow forgetting.** Classical definitions of catastrophic forgetting focus on output degradation (*shallow forgetting*). More recent studies show that internal representations often retain past-task structure, recoverable via probes (*deep forgetting*) (Murata et al., 2020; Ramasesh et al., 2020; Fini et al., 2022; Davari et al., 2022; Zhang et al., 2022; Hess et al., 2023). Replay is known to mitigate deep forgetting in hidden layers (Murata et al., 2020; Zhang et al., 2022). To our knowledge, we are the first to demonstrate that deep and shallow forgetting scale fundamentally differently with buffer size.

**Neural Collapse.** NC describes the emergence of an ETF structure in last-layer features at convergence (Papyan et al., 2020; Mixon et al., 2022; Tirer & Bruna, 2022; Jacot et al., 2024; Súkeník et al., 2025). Extensions address class imbalance (*Minority Collapse*) (Fang et al., 2021; Dang et al., 2023; Hong & Ling, 2023) and overcomplete regimes (Jiang et al., 2024; Liu et al., 2023; Wu & Papyan, 2024). In continual learning, NC has been leveraged to fix global ETF heads to reduce forgetting (Yang et al., 2023; Dang et al., 2024; Wang et al., 2025). Our approach is distinct: we apply NC theory to the *asymptotic analysis* of continual learning and introduce the multi-head setting, common in CL but previously unexplored in NC theory.

**OOD detection.** Early work observed that OOD inputs yield lower softmax confidence (Hendrycks & Gimpel, 2018), while later studies showed that OOD features collapse toward the origin due to low-rank compression (Kang et al., 2024). Recent results connect this behavior to NC: $L_2$ regularization accelerates NC and sharpens ID/OOD separation (Haas et al., 2023), and ID/OOD orthogonality has been proposed as an additional NC property, with OOD scores derived from ETF subspace norms (Ammar et al., 2024). Our work extends these insights by formally establishing orthogonality, clarifying the role of weight decay and feature norms, and—crucially—providing the first explicit link between OOD detection and forgetting in CL.

## 6 FINAL DISCUSSION & CONCLUSION

**Takeaways.** This work has shown that: (1) replay affects network features and classifier heads in fundamentally different ways, leading to a slow reduction of the replay efficiency gap as buffer size increases; (2) the Neural Collapse framework can be systematically extended to continual learning, with particular emphasis on the multi-head setting—a case not previously addressed in the NC literature; (3) continual learning can be formally connected to the out-of-distribution (OOD) detection literature, and our results extend existing discussions of NC on OOD data. We further elucidated how weight decay and the growth of class feature norms jointly determine linear separability in feature space. Our analysis also uncovered several unexpected phenomena: (i) class feature norms grow with the number of classes in class- and task-incremental learning; (ii) multi-head models yield structurally lower-rank feature spaces compared to single-head models; and (iii) weight decay exerts a double-edged influence on feature separability, with its effect differing across continual learning setups.

**Limitations.** Our theoretical analysis adopts an asymptotic perspective, thereby neglecting the transient dynamics of early training, which are likely central to the onset of forgetting (Łapacz et al., 2024). Moreover, our modeling of replay buffers as interpolations between idealized extremes simplifies the true distributional dynamics and may not fully capture practical scenarios. Finally, many aspects of feature-space evolution under sequential training—particularly the nature of cross-task interactions in multi-head architectures—remain poorly understood and require further investigation.

**Broader Implications.** By establishing a formal link between Neural Collapse, OOD representations, and continual learning, our findings highlight key design choices—including buffer size, weight decay, and head structure—that shape the stability of past-task knowledge. These results raise broader questions: *What constitutes an "optimal" representation for continual learning? Is the Neural Collapse structure beneficial or detrimental in this context?* Our results suggest that while NC enhances feature organization, it also exacerbates the mismatch between replay and true distributions, thereby contributing to the replay efficiency gap. Addressing these open questions will be essential for designing future continual learning systems.

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

# Appendix

## Table of Contents

# A EMPIRICAL APPENDIX

## A.1 EXPERIMENTAL EETAILS

We utilize the benchmark codebase developed by Buzzega et al. (2020)[2]. To accommodate our experiments we performed several changes to the default implementation.

**Training configurations.** Table 1 summarizes the configurations used in our main experiments. All models are trained in an offline continual learning setting, where each task's dataset is trained for a specified number of iterations before transitioning to the next task. Models are trained to reach error convergence on each task and more training does not improve performance. For all experiments, the random seeds were set to [1000, 2000, 3000].

We define the number of tasks as specified in Table 1. The classes of each dataset are then distributed across these tasks. In the DIL setting, the same class labels are reused for every task. The class ordering was randomized in each run, meaning that a specific tasks consist of different classes in each run. This was done to ensure that the results are not biased by a specific class sequence. However, we observed that this increases the variance when metrics are evaluated task wise compared to using a fixed class assignment.

| Dataset | Tasks | Epochs first task | Network | Batch Size |
|---------|-------|-------------------|---------|------------|
| Cifar100 | 10 | 200 | ResNet18 (11M) | 64 |
| Cifar100 | 10 | 40 | ViT base, pretrained on ImageNET (86M) | 256 |
| TinyIMG | 10 | 200 | ResNet18 (11M) | 64 |
| CUB200 | 10 | 80 | ResNet50, pretrained on ImageNET (24M) | 32 |

Table 1: **Experiment configurations.**

**Hyper parameters** Our hyper parameters were largely adapted from (Buzzega et al., 2020) and are listed in Table 2. We use a constant learning rate. For all buffer sizes the same hyper parameters are used. Finally to study the effects of weight decay, we vary the weight decay strength in our experiments while keeping all other factors constant.

| Dataset | Method | Optimizer | Hyper Parameters |
|---------|--------|-----------|------------------|
| Cifar100, ResNet | ER | SGD | $lr : 0.1, wd : 0.0001$ |
| Cifar100, ResNet | DER | SGD | $lr : 0.03, \alpha = 0.3,$ |
| Cifar100, ResNet | FDR | SGD | $lr : 0.03, \alpha = 0.3,$ |
| Cifar100, ResNet | iCaRL | SGD | $lr : 0.1, wd : 0.00005$ |
| Cifar100, ViT | ER | AdamW | $lr : 0.0001, wd = 0.0001$ |
| TinyIMG, ResNet | ER | SGD | $lr : 0.1, wd : 0.0001$ |
| CUB200, ResNet | ER | SGD | $lr : 0.03, wd : 0.0001$ |

Table 2: **Hyper parameters**

**Datasets and preprocessing.** We adopt publicly available image classification benchmarks: Cifar100 ($32 \times 32$ RGB, 50000 samples across 100 classes), TinyIMG ($64 \times 64$ RGB across 100000 samples, 200 classes) and CUB200 ($224 \times 224$ RGB, 12000 samples across 200 classes). Standard train/test splits are used. We apply standard augmentations like random crops and flips, without increasing the dataset size.

**Experience Replay (ER).** In our implementation of ER, we adopt a balanced sampling strategy in which each task contributes equally to the mini-batches. While this strategy would normally require

---

[2]Their codebase is publicaly availible at: https://github.com/aimagelab/mammoth

more iterations for later tasks, we avoid this by fixing the total number of iterations across all tasks to match those performed on the first task—effectively reducing the number of epochs for later tasks.

To maintain precise control over the buffer composition, we employ an offline sampling scheme to populate the buffer. Samples (together with their labels) from a task are added to the buffer only after training on that task is completed. This guarantees a balanced number of stored samples per task and class. Because the buffer size is specified as a percentage of the task-wise dataset, the number of stored samples per task remains constant and equal to the chosen buffer percentage. As more tasks are encountered, these fixed per-task allocations accumulate, resulting in a steadily increasing overall buffer size. Table 3 specifies the buffer sizes used in the experiments. When setting the buffer size to 0, ER naturally reduces to standard SGD.

| Buffer Sizes (% of task-wise dataset) |
| --- |
| 0, 1, 2, 3, 4, 5, 6, 8, 10, 100 |

Table 3: Buffer sizes expressed as percentages of each task-wise dataset. The same values are used consistently across all experimental configurations.

**Measures of superficial and deep forgetting.** *Shallow forgetting* quantifies the drop in output accuracy on past tasks after learning new ones, defined as

$$F_{i \to j}^{\text{shallow}} = A_{jj} - A_{ij},$$

where $A_{ij}$ is the accuracy on task $j$ measured after learning session $i$.

*Deep forgetting* measures the loss of discriminative information in the features themselves, independent of the head. To measure it, we train a logistic regression classifier (scikit-learn's LogisticRegression, default settings, C=10) on frozen features extracted from the full dataset after learning session $i$. The resulting accuracy, evaluated at the end of session $j$, is denoted by $A_{ij}^\star$. Formally,

$$F_{i \to j}^{\text{deep}} = A_{jj}^\star - A_{ij}^\star.$$

For single-head models, one probe is trained over all classes; for multi-head architectures, one probe per task-specific head is used.

### A.2  Figure details

This subsection details the computations behind the figures presented in the main text.

**Figure 2.** Forgetting metrics are evaluated after the final training session, following the procedure described in Appendix A.1, and across buffer sizes specified in Table 3. Different line styles correspond to distinct continual learning settings (TIL, DIL, CIL).

**Figure 3.** Neural Collapse (NC) metrics are computed for each task every 100 steps during training of a ResNet from scratch on CIFAR100 in both CIL, DIL and TIL settings. Metrics are evaluated on the available training data, which includes the current task's dataset plus the replay buffer which contains 5% of the past tasks' dataset. In TIL, for $\mathcal{NC}2$ the within-class-pair values for each task are shown in the standard task colors, while the values across class pairs from different tasks are highlighted in violet. The brown vertical lines indicate the task switches.

Note that in DIL each task contains the same set of classes. If class means were computed naively across tasks, differences specific to each task would be obscured, preventing us from identifying task-wise trends. Therefore Figure 3 evaluates the NC metrics separately for each task. In contrast Figure 15 reports the NC metrics computed jointly over all tasks in DIL. Importantly, both approaches produce consistent results.

**Figure 4.** Norm of $\tilde{\mu}_c(t)$ projected to $S_t$, averaged over all classes belonging to a task, is computed every 100 steps during training of a ResNet from scratch on CIFAR100 under CIL. The brown vertical lines indicate the task switches. In DIL $\tilde{\mu}_c(t)$ is computed separately for every task.

**Figure 5** Measurements are collected after the final training session on Cifar100 using a ResNet trained from scratch, averaged over all past-task classes. In DIL we again calculate the class means

task wise. The buffer sizes correspond to those listed in Table 3. The signal-to-noise ratio (SNR) is computed as described in Section 1.1. The second panel displays the normalized variance ratio, where the *within-class variance* is defined as $\frac{1}{|\mathcal{C}|} \sum_{c \in \mathcal{C}} \mathrm{Tr}(\mathrm{Cov}(\phi(x) \mid x \in X_c))$, and the *between-class variance* is defined as $\mathrm{Tr}(\mathrm{Cov}(\{\mu_c\}_{c \in \mathcal{C}}))$, with $\mu_c$ the population feature mean vector of class $c$. The third panel displays the average ratio: $\|S_t^\perp \tilde{\mu}_c(t)\| / \|S_t \tilde{\mu}_c(t)\|$. And the fourth panel displays the average norm of $\tilde{\mu}_c(t)$ and $\hat{\tilde{\mu}}_c(t)$.

**Figure 6 (first three plots).** At the end of the last training session, the network is evaluated on multiple datasets under a CIL protocol. For CUB200 we do not report buffer sizes which are smaller then 4%, as at least two samples per class are needed to calculate the covariance matrix. For each buffer size reported in Table 3, we collect the class-wise mean vectors $\hat{\mu}_c(t)$ and covariances $\hat{\Sigma}_c(t)$ from the buffer, as well as the corresponding population statistics $\mu_c(t)$ and $\Sigma_c(t)$. The following metrics are computed and averaged across past classes:

- Mean gap: $\|\mu_c(t) - \hat{\mu}_c(t)\|_2$.

- Covariance gap: $\|\Sigma_c(t) - \hat{\Sigma}_c(t)\|_F$, the Frobenius norm of the difference between covariances.

- Covariance rank: both the rank of the population covariance matrix $\Sigma_c(t)$ and the observed covariance matrix $\hat{\Sigma}_c(t)$ are reported. Note that the rank is upper bounded by the number of samples which are used to calculate the covariance matrix.

These quantities quantify the discrepancy between the buffer and true class distributions in feature space, which drives shallow forgetting.

**Figure 6 (right-most panel).** Same experimental setup as the first three panels. We evaluate different linear classifiers on TinyIMG using class-wise feature statistics. Specifically, we construct linear discriminant analysis (LDA) classifiers. For a class $c$, the LDA decision rule is

$$\hat{y}(x) = \arg\max_c \; (x - \hat{\mu}_c(t))^\top \hat{\Sigma}^{-1}(t)(x - \hat{\mu}_c(t)),$$

where $\hat{\mu}_c$ denote the estimated class mean and $\hat{\Sigma}$ is a shared covariance matrix. We vary the estimates used for each class as follows:

- **Full population:** both mean $\mu_c(t)$ and covariance $\Sigma(t)$ are measured on population, $\Sigma(t)$ is pooled across all classes.

- **Population means (ID cov):** mean is taken from population $\mu_c(t)$, but covariance is fixed to the identity.

- **Observed means (ID cov):** mean is taken from observed samples $\hat{\mu}_c(t)$, but covariance is fixed to the identity.

- **Full buffer:** both mean $\hat{\mu}_c(t)$ and covariance $\hat{\Sigma}(t)$ are computed from the observed samples, $\hat{\Sigma}(t)$ is pooled across all classes.

This evaluation highlights how errors in buffer-based mean and covariance estimates contribute to shallow forgetting, and quantifies the impact of each component on linear decoding performance.

## A.3   ABLATIONS

### A.3.1   EFFECT OF PRETRAINING

Our results in Figure 3 demonstrate that models trained from scratch indeed undergo Neural Collapse (NC) in a continual learning setting. However, when comparing this to pre-trained models, we find that while both settings converge to the same asymptotic feature geometry, the pre-trained models do so at a substantially accelerated rate. This difference in convergence speed is illustrated in Figure 7. A side-by-side comparison of the initial 1500 iterations confirms that pre-trained models rapidly achieve the high NC scores that their de novo counterparts only reach much later in training.

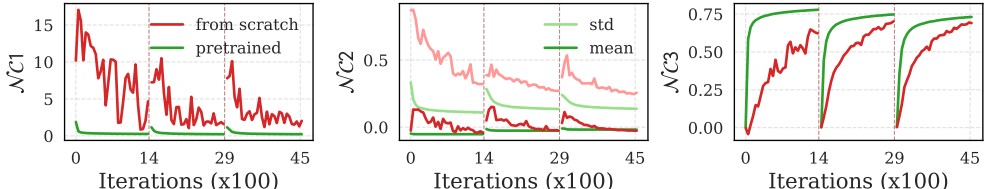

Figure 7: Convergence to Neural Collapse (NC) for pre-trained versus from-scratch models under CIL on CUB200. Pre-trained models achieve asymptotic NC scores significantly faster than their de novo counterparts.

### A.3.2 FEATURE BOTTLENECK: WHEN $d \ll K$

In the main paper, we considered settings where the feature dimension exceeds the number of classes. However, in many practical applications, such as language modeling, the number of classes (e.g., vocabulary size) is typically much larger than the feature dimension. Recent work by (Liu et al., 2023) explored this regime. To examine how our framework behaves under these conditions, we conducted additional experiments by modifying the Cifar100 with ResNet setup. Specifically, we split Cifar100 into four tasks of 25 classes each and inserted a bottleneck layer of dimension 10 between the feature layer and the classifier head. All other components are left unchanged.

As illustrated in Figure 8, variability collapse $\mathcal{NC}1$ and neural duality $\mathcal{NC}3$ remain in this constrained setting. However, the equiangularity $\mathcal{NC}2$ exhibits significant degradation. While the mean pairwise cosine similarity aligns with theoretical expectations, its standard deviation increases substantially to $\approx 0.3$ compared to the typical convergence levels of $\approx 0.1$ in our standard setting (Figure 12 and similarly observed by Papyan et al. (2020)). Therefore, even though the mean appears correct, the underlying structure is not, as the standard deviation is far to high. This high variance indicates a failure to converge to a rigid simplex, suggesting that in the $d \ll K$ regime alternative geometric structures must be considered, such as the Hyperspherical Uniformity explored by Liu et al. (2023).

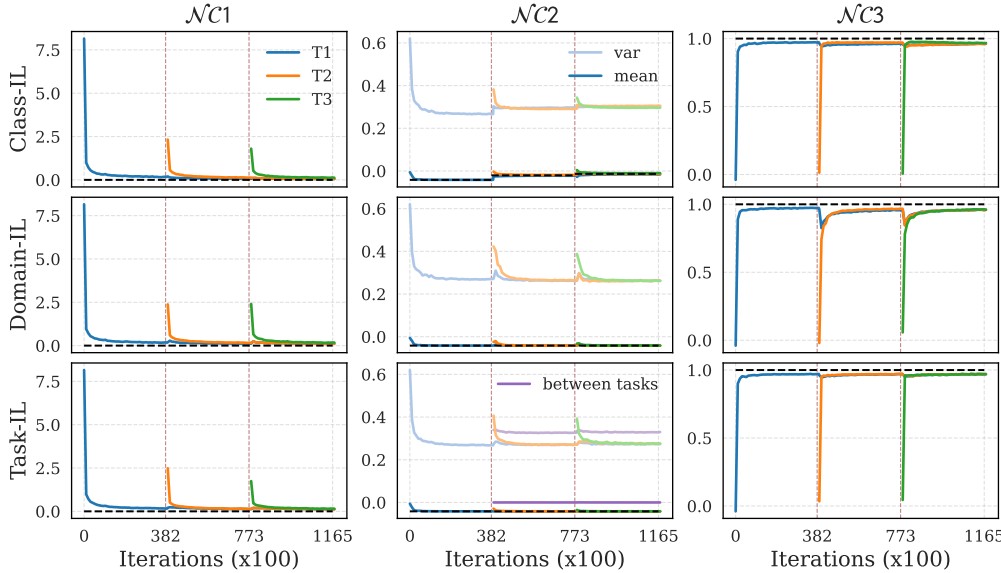

Figure 8: *NC metrics in the bottleneck regime (d=10)*. Same setup as Figure 3. Results for Cifar100 (4 tasks, 25 classes, 5% replay). While variability collapse (NC1) and duality (NC3) persist, the rigid ETF structure (NC2) degrades, exhibiting high variance in pairwise angles.

Crucially, we find that the **replay efficiency gap persists** (Figure 8) despite this geometric shift. This implies that the decoupling between feature separability and classifier alignment is not contingent on

the specific ETF geometry. Rather, the gap is a fundamental phenomenon that emerges even when the learned representations follow alternative geometric structures, provided they remain collapsed.

| Dataset | Learning paradigm | Shallow forgetting | Deep forgetting |
|---------|-------------------|-------------------|-----------------|
| Cifar100 | CIL | $52.27 \pm 2.41$ | $26.15 \pm 1.92$ |
| Cifar100 | DIL | $46.38 \pm 1.88$ | $35.07 \pm 0.51$ |
| Cifar100 | TIL | $18.45 \pm 2.78$ | $14.89 \pm 1.77$ |

Table 4: The deep-shallow forgetting gap persists in the low feature-dimension regime (Cifar100, ResNet with 5% replay).

### A.3.3 Effect of head initialization

We analyze the empirical evolution of the centered class-mean norm $\beta_t$ across tasks. As illustrated in Figure 9, we observe a distinct architectural split: $\beta_t$ increases monotonically in setups with increasing number of classes (CIL and TIL), whereas it remains asymptotically stable in DIL.

We attribute this drift to a *weight norm asymmetry* induced by the sequential expansion of the network outputs. In CIL and TIL, new head weights are typically instantiated using standard schemes (e.g., Kaiming Uniform), which initialize weights with significantly lower norms than those of the already-converged heads from previous tasks. This creates a recurrent initialization shock. In contrast, DIL employs a fixed, shared head across all tasks, inherently avoiding this discontinuity.

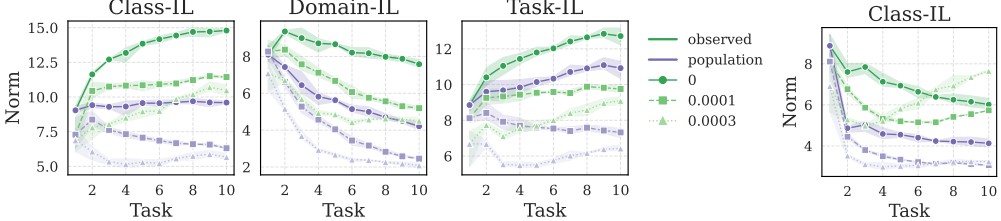

Figure 9: Average (over all seen classes $c$) norm of the centered observed class means $\tilde{\mu}_c(t)$ and population class means $\tilde{\mu}_c(t)$ after training each task on Cifar100, with varying weight decay coefficients. The three panels on the left correspond to the default head initialization, which results in a progressively increasing norm in both CIL and TIL. The rightmost panel shows the results when each new head is initialized with the same norm as the previously trained heads. This adjustment prevents the norm from growing.

To validate this hypothesis, we performed an ablation using a *norm-matching initialization* strategy. In this setup, the weights of new tasks are scaled to match the average norm of existing heads while preserving their random orientation.

Results in Figure 9 (right) confirm that this intervention effectively suppresses the progressive growth of $\beta_t$, recovering the stationary norm behavior observed in DIL. Interestingly, while this adjustment stabilizes the geometric scale of the representation, we found it yields negligible impact on final forgetting or test accuracy metrics.

### A.4 Additional figures and empirical substantiation

This subsection includes placeholder figures for concepts discussed in the main text, for which specific existing figures were not available or suitable for direct inclusion in the main body.

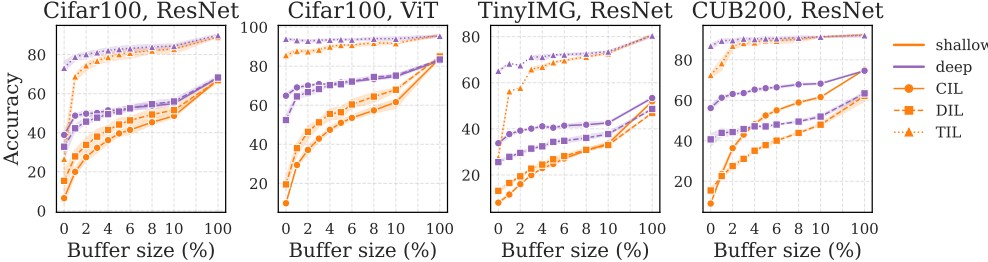

Figure 10: Same setup as Figure 2. This plot reports test accuracy.

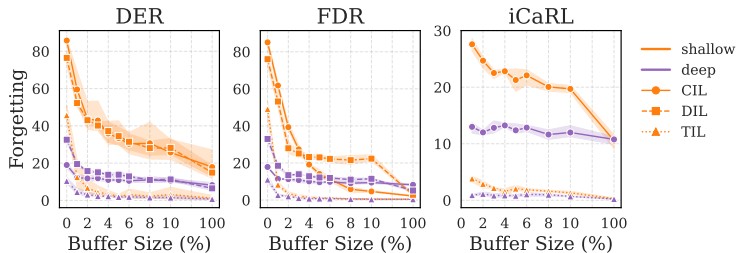

Figure 11: Deep–shallow forgetting gap for *Dark Experience Replay* (DER), *Functional Distance Relation* (FDR) and *Incremental Classifier and Representation Learning* (iCaRL) on Cifar100 with ResNet. Note that iCaRL does not support DIL nor training without buffer.

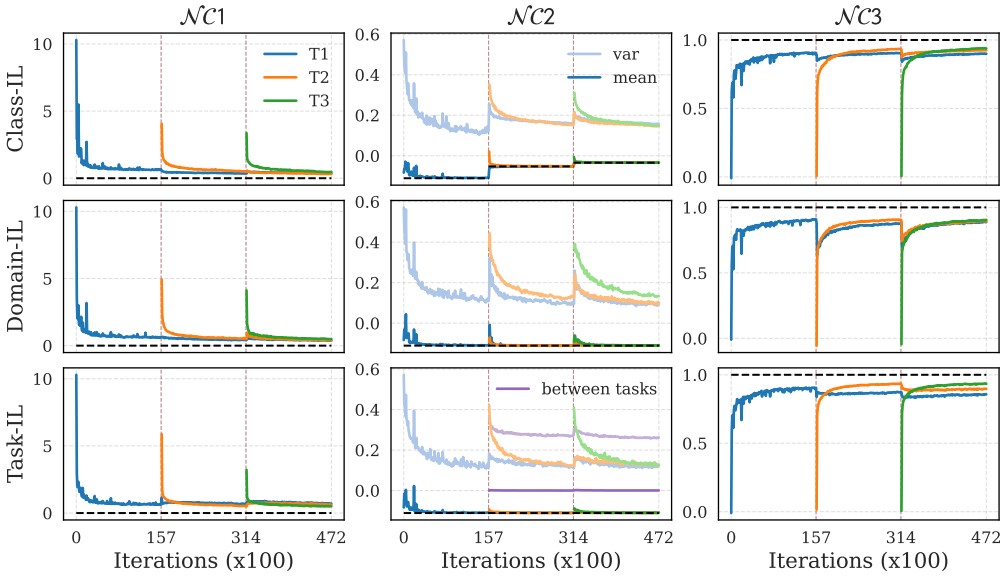

Figure 12: Same setup as Figure 3. This plot shows the NC metrics on Cifar100 with 5% replay. For NC2, both the mean and standard deviation are shown.

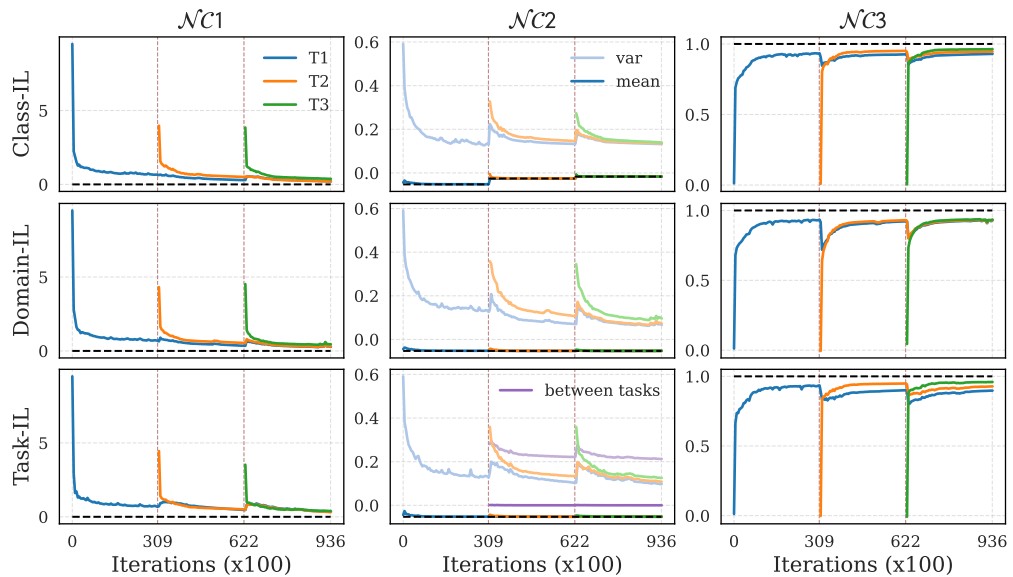

Figure 13: Same setup as Figure 3. This plot shows the NC metrics on TinyIMG with 5% replay. For NC2, both the mean and standard deviation are shown.

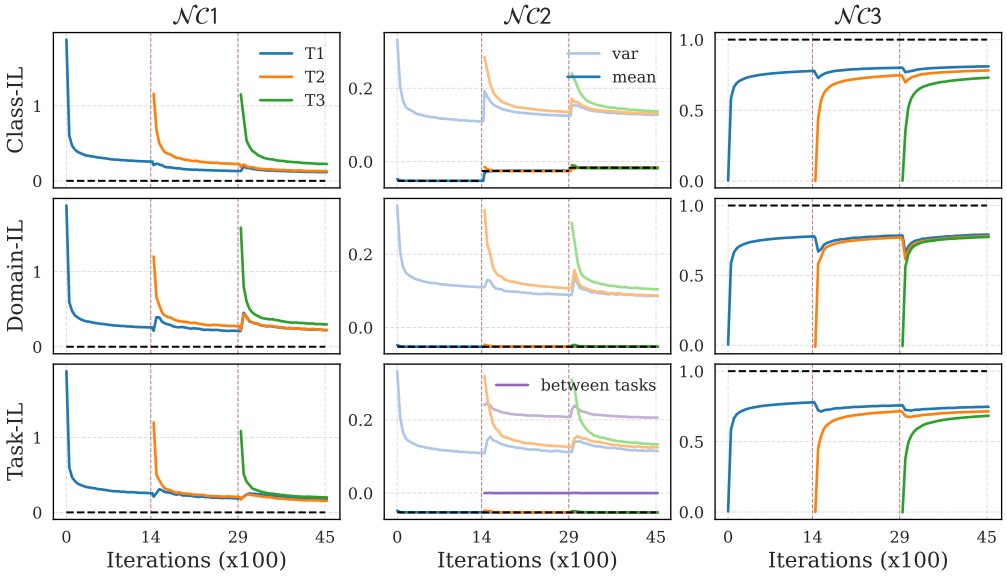

Figure 14: Same setup as Figure 3. This plot shows the NC metrics on CUB200 with 10% replay. For NC2, both the mean and standard deviation are shown.

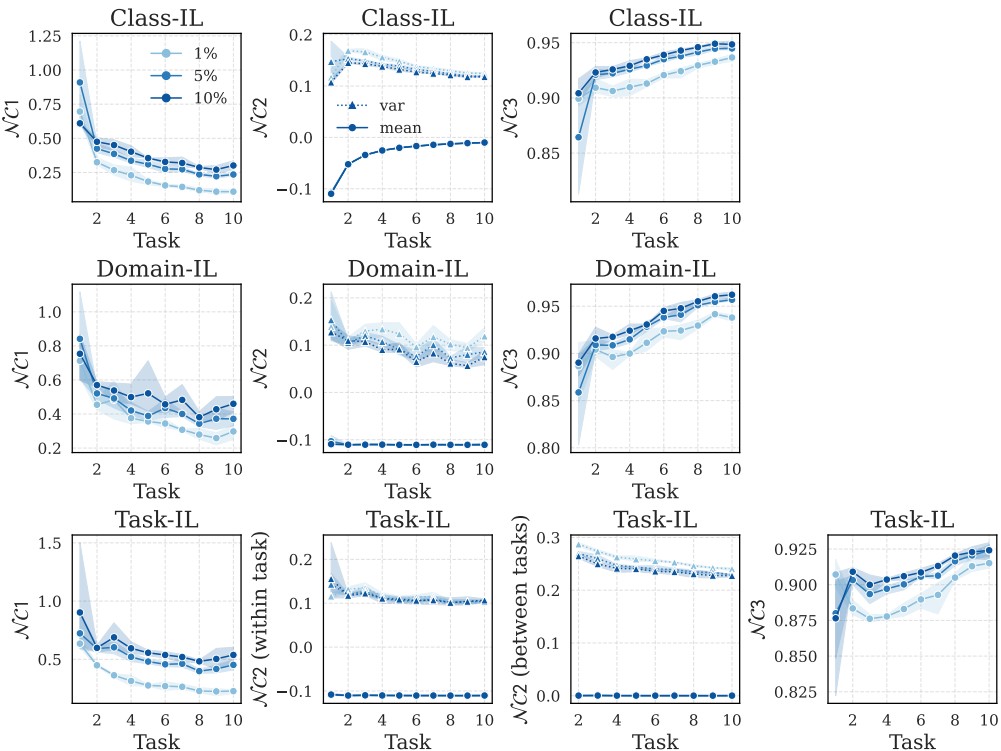

Figure 15: Evolution of Neural Collapse metrics over all tasks in sequential training (Cifar100, ResNet) varying the replay buffer size. Neural Collapse is stronger for smaller buffers.

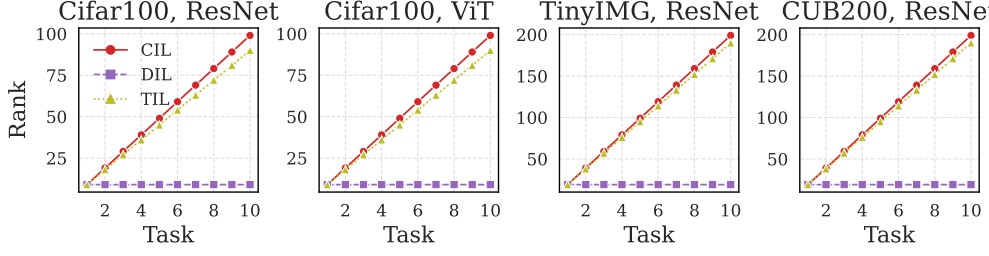

Figure 16: Rank of the centered observed class mean matrix $\tilde{\tilde{U}}(t)$. In CIL and TIL the rank increases (at different speeds) as more tasks are learned, whereas in DIL the rank remains remains constant.

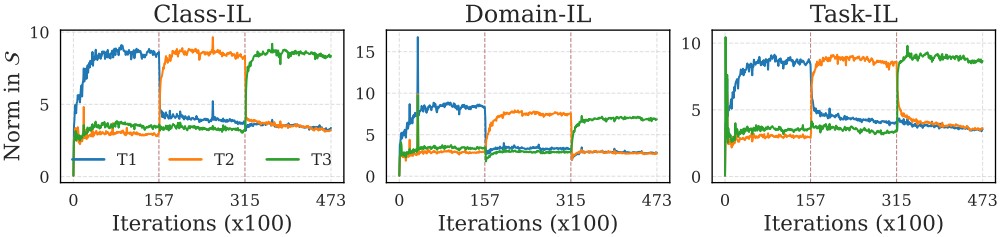

Figure 17: Same setup as Figure 4. This plot shows the average norm of $\tilde{\mu}_c(t)$ when projected to $S_t$ for CIL, DIL and TIL on Cifar100 with no replay.

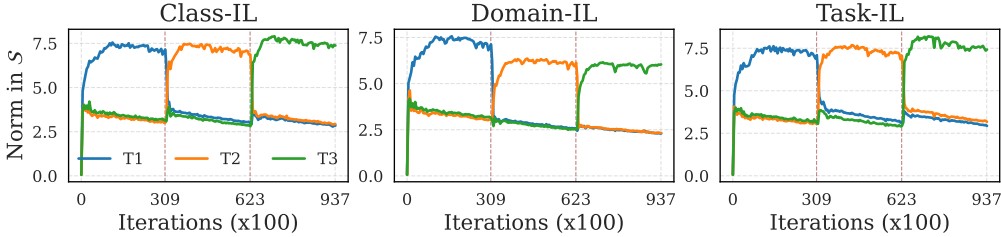

Figure 18: Same setup as Figure 4. This plot shows the average norm of $\tilde{\mu}_c(t)$ when projected to $S_t$ for CIL, DIL and TIL on TinyIMG with no replay.

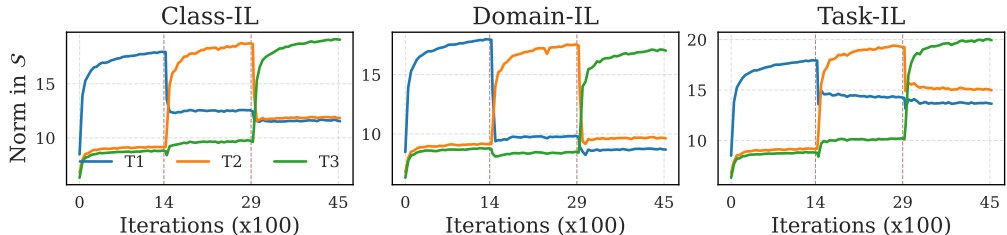

Figure 19: Same setup as Figure 4. This plot shows the average norm of $\tilde{\mu}_c(t)$ when projected to $S_t$ for CIL, DIL and TIL on CUB200 with no replay.

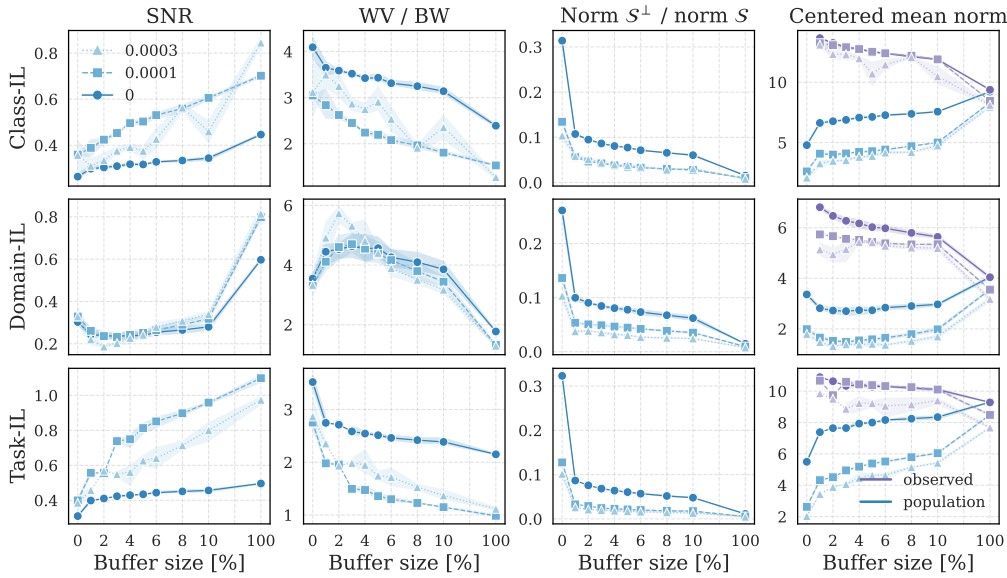

Figure 20: Same setup as Figure 5. This plot displays the results for TinyIMG.

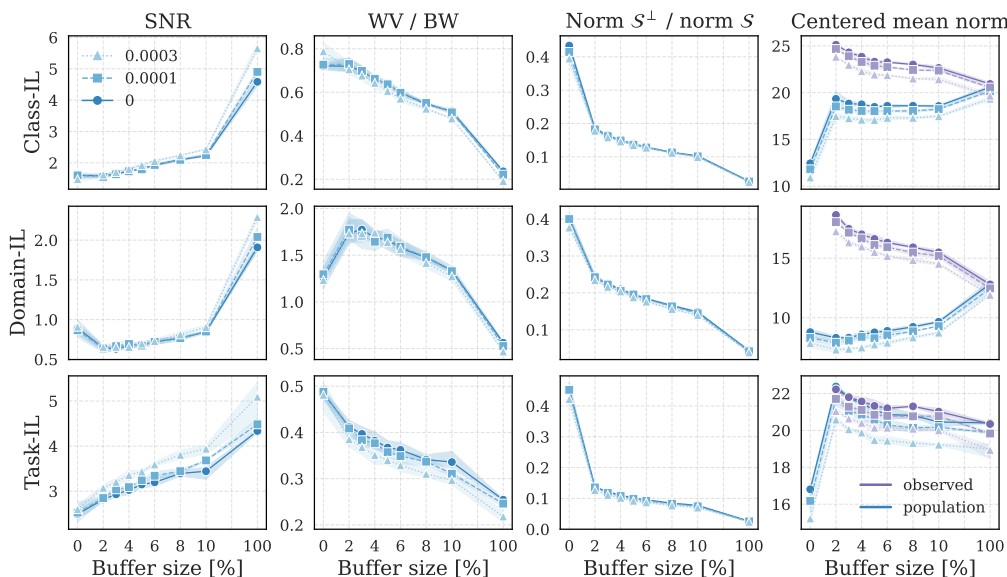

Figure 21: Same setup as Figure 5. This plot displays the results for CUB200.

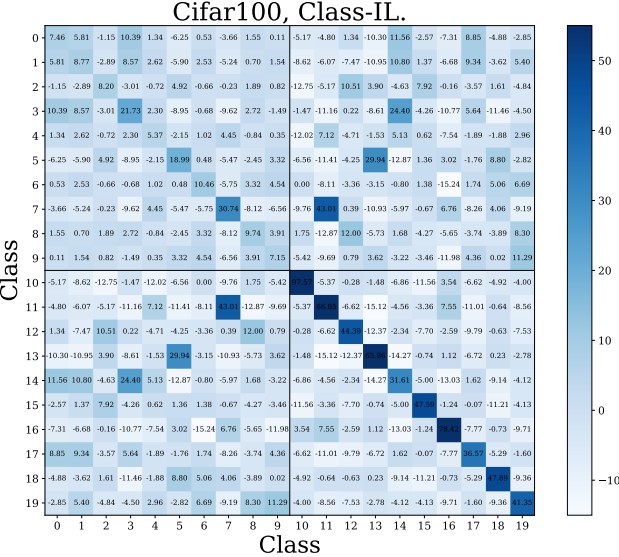

Figure 22: Class-wise inner products of the centered population class means $\tilde{\mu}_c(t)$ on Cifar100 under CIL after the second task. Classes 0 to 9 belong to the first task, while 10 to 19 belong to the second task. The classes belonging to task 2 are structured according to the NC regime, while classes belonging to task 1 show no structure.

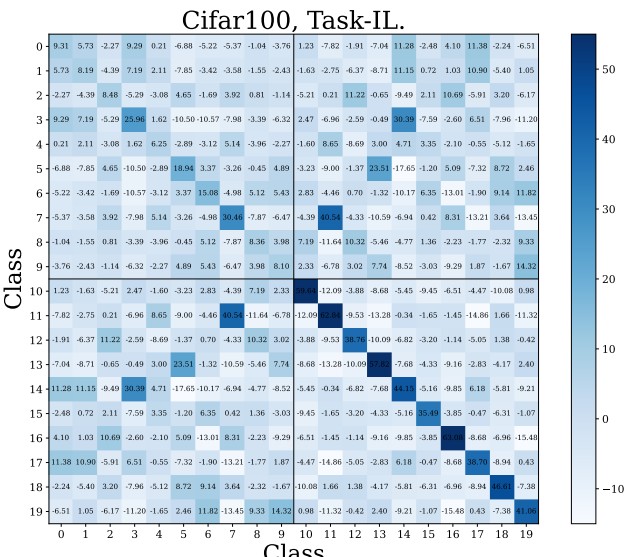

Figure 23: Class-wise inner products of the centered population class means $\tilde{\mu}_c(t)$ on Cifar100 under TIL after the second task. Classes 0 to 9 belong to the first task, while 10 to 19 belong to the second task. The classes belonging to task 2 are structured according to the NC regime, while classes belonging to task 1 show no structure.

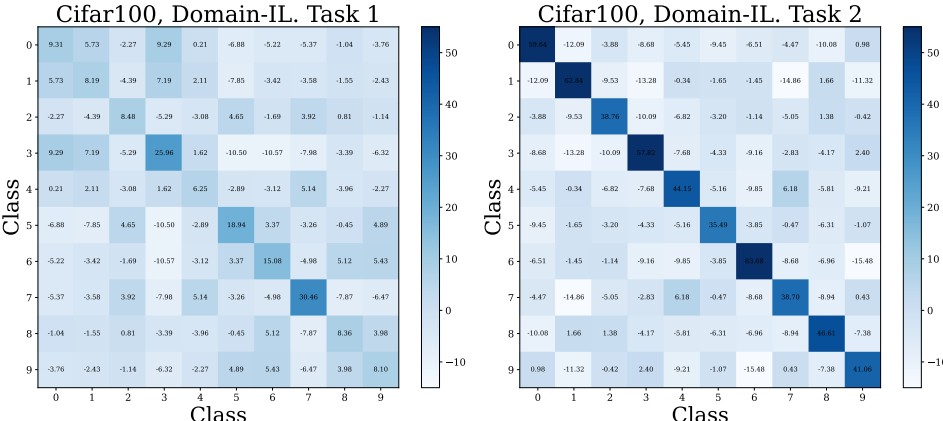

Figure 24: Class-wise inner products of the centered population class means $\tilde{\mu}_c(t)$ on Cifar100 under DIL after the second task. The left plot shows the results for samples belonging to the first task, while the right plot shows results for samples from the second task. Task 2 is structured according to the NC regime, while task 1 shows no structure.

# B    OVERVIEW OF RELATED WORK

Our work intersects several strands of research. First, it builds on the literature studying the geometric structures that emerge in neural feature spaces, extending these analyses to the sequential setting of continual learning and accounting for the additional challenges introduced by different head expansion mechanisms. Second, it connects to the out-of-distribution detection literature, where we reinterpret forgetting as feature drift and broaden existing insights to a more general framework. Finally, it contributes to the continual learning literature that disentangles knowledge retention at the representation level from that at the output level, highlighting the systematic mismatch between the two in replay.

**Deep and shallow forgetting**    Traditionally, *catastrophic forgetting* is defined as the decline in a network's performance on a previously learned task after training on a new one, with performance measured at the level of the network's outputs. We refer to this notion as *shallow forgetting*. In contrast, Murata et al. (2020) highlighted that forgetting can also be assessed in terms of the network's internal representations. They proposed quantifying forgetting at a hidden layer $l$ by retraining the subsequent layers $l+1$ to $L$ on past data and comparing the resulting accuracy to that of the original network. Applied to last-layer features, this procedure coincides with the widely used *linear probe* evaluation from the representation learning literature, often complemented by kNN estimators, to assess task knowledge independently of a task-specific head. In this work, we refer to the loss of information at the feature level as *deep forgetting*. This probing-based approach has also been adopted in continual learning studies (Ramasesh et al., 2020; Fini et al., 2022). Multiple works have since reported a consistent discrepancy between deep and shallow forgetting across diverse settings (Davari et al., 2022; Zhang et al., 2022; Hess et al., 2023). Of particular relevance to our study are the findings of Murata et al. (2020) and Zhang et al. (2022), who observed that replay methods help mitigate deep forgetting in hidden representations. To our knowledge, however, we are the first to demonstrate that deep and shallow forgetting exhibit categorically different scaling behaviors with respect to replay buffer size.

**Neural Collapse and continual learning**    Neural Collapse (NC) was first introduced by Papyan et al. (2020) to describe the emergence of a highly structured geometry in neural feature spaces, namely a *simplex equiangular tight frame* (simplex ETF) characterized by the NC1–NC4 properties. Its optimality for neural classifiers, as well as its emergence under gradient descent, was initially established under the simplifying *unconstrained feature model* (UFM) (Mixon et al., 2022). Subsequent theoretical work extended these results to end-to-end training of modern architectures with both MSE and CE loss on standard classification tasks (Tirer & Bruna, 2022; Jacot et al., 2024; Súkeník et al., 2025). Generalizations of NC have been proposed for settings where the number of classes exceeds the feature dimension, precluding a simplex structure. In such cases, the NC2 and NC3 properties are extended via one-vs-all margins (Jiang et al., 2024) or hyperspherical uniformity principles (Liu et al., 2023; Wu & Papyan, 2024). Another important line of work concerns the class-imbalanced regime, which arises systematically in continual learning. Here, the phenomenon of *Minority Collapse* (MC) (Fang et al., 2021) has been observed, in which minority-class features are pushed toward the origin. Dang et al. (2023); Hong & Ling (2023) derived an exact law for this collapse, including a threshold on the number of samples below which features collapse to the origin and above which the NC configuration is gradually restored. Because class imbalance is inherent in class-incremental continual learning, NC principles have also been leveraged to design better heads. Several works (Yang et al., 2023; Dang et al., 2024; Wang et al., 2025) impose a fixed global ETF structure in the classifier head, rather than learning it, to mitigate catastrophic forgetting. To our knowledge, we are the first to use NC theory to analyze the *asymptotic* geometry of neural feature spaces in continual learning. In doing so, we introduce the multi-head setting, which is widely used in continual learning but has not been formally studied in the NC literature. While a full theory of multi-head NC lies beyond the scope of this paper, our empirical evidence provides the first steps toward such a framework.

**Out-Of-Distribution Detection**    Out-of-distribution (OOD) detection is a critical challenge for neural networks deployed in error-sensitive settings. Hendrycks & Gimpel (2018) first observed that networks consistently assign lower prediction confidences to OOD samples across a wide range of tasks. Subsequent work has shown that OOD samples occupy distinct regions of the represen-

tation space, often collapsing toward the origin due to the in-distribution filtering effect induced by low-rank structures in the backbone (Kang et al., 2024). Haas et al. (2023) connected this phenomenon to Neural Collapse (NC), demonstrating that $L_2$ regularization accelerates the emergence of NC and sharpens OOD separation. Building on this, Ammar et al. (2024) proposed an additional NC property—*ID/OOD Orthogonality*—which postulates that in-distribution and out-of-distribution features become asymptotically orthogonal. They further introduced a detection score based on the norm of samples projected onto the simplex ETF subspace $S$, which closely parallels the analysis in our work. Our results extend this line of research by providing formal evidence for the ID/OOD orthogonality hypothesis, offering a precise characterization of the roles of weight decay and feature norm, and, to our knowledge, establishing the first explicit connection between catastrophic forgetting and OOD detection.

## C   MATHEMATICAL DERIVATIONS

**Notation**

| | |
|---|---|
| $D_n$ | Dataset of task $n$ |
| $\hat{D}_n$ | Training dataset during session $n$ -may include buffer |
| $\bar{D}$ | Datasets of all tasks combined |
| $X_c$ | Instances of class $c$ in all tasks |
| $\mathcal{L}(\theta, D)$ | Average loss function over $D$ |
| $\lambda$ | Weight decay factor |
| $\eta$ | SGD learning rage |
| $f_\theta$ | Network function $\mathbb{R}^{d_1} \to \mathbb{R}^P$ |
| $\phi$ | Feature map $\mathbb{R}^{d_1} \to \mathbb{R}^{d_L}$ |
| $h$ | Network head $\mathbb{R}^{d_L} \to \mathbb{R}^P$ |
| $W_h$ | Network head weights |
| $W_h^n$ | Network head weights for classes of task $n$ (only multi-head) |
| $\mu, \Sigma, \sigma^2$ | The mean, covariance and variance of a distribution |
| $\tilde{\mu}_c = \mu_c - \mathbb{E}_c[\mu_c]$ | Centered class $c$ mean |
| $S = \mathrm{span}(\tilde{\mu}_1, \ldots, \tilde{\mu}_K)$ | Centered mean span |
| $\tilde{U} = [\tilde{\mu}_1, \ldots, \tilde{\mu}_K]$ | Centered mean matrix |
| $P_A$ | Projection onto the space $A$ |
| $\beta_t = \|\mu(t)\|^2$ | Training class (squared) norm |

### C.1   SETUP

Consider a neural network with weights $\theta$, divided into a non-linear map $\phi : \mathbb{R}^{d_1} \to \mathbb{R}^{d_L}$ and a linear head $h : \mathbb{R}^{d_L} \to \mathbb{R}^K$. The function takes the form:

$$f_\theta(x) = W_h \, \phi(x) \, + b_h$$

We hereafter refer the map $\phi(x)$ as *features* or *representation of the input* $x$, and to $f(x)$ as output. The network is trained to minimize a classification loss $\ell((x, y), f_\theta)$ on a given dataset $D$. We denote by $\mathcal{L}(D, \theta)$ the average $\ell((x, y), f_\theta)$ over $D$, and where clear we leave $D$ implicit. The loss is assumed to be convex in the network output $f_\theta(\cdot)$.

For each task $n$ a new dataset $D_n$ is provided, with $K$ classes. We denote by $\bar{D}_t = \cup_{n \leq t} D_n$ the union of all datasets for tasks 1 to $t$ and simply $\bar{D}$ the union of all datasets across all tasks. Moreover, we denote by $\hat{D}_t$ the training data used during the session $t$ - which may include a buffer. For a given class $c$ we denote by $X_{D,c}$ the available inputs from that class, i.e. $X_{D,c} = \{x : (x, y) \in D \text{ and } y = c\}$. We use $X_c = X_{\bar{D},c}$ the set of inputs for class $c$ across all learning sessions. We assume the number of classes $K$ to be predicted to be the same for each task.

For a given class data $X_c$ the *class mean feature* vector is:

$$\mu_c(\bar{D}) = \mathbb{E}_{X_c} [\phi(x)].$$

We call $\mu_c(\bar{D})$ the *population* mean, to distinguish it from the *buffer* mean $\hat{\mu}_c(B)$. If a given class appears in multiple training session, we additionally distinguish between $\mu_c(\bar{D})$ and $\hat{\mu}_c(\hat{D}_t)$, where the latter is the *observed* mean. For a set of classes $\{1, \ldots, K\}$ in a dataset $D$ the *global mean feature* vector is:

$$\mu_G(\bar{D}) = \mathbb{E}_c \, \mathbb{E}_{X_c} [\phi(x)] = \mathbb{E}_{\bar{D}} [\phi(x)],$$

which we call *population* global mean to distinguish it from the *buffer* global mean $\hat{\mu}_G(B)$. Finally, the *centered class mean feature* vector is:

$$\tilde{\mu}_c(\bar{D}) = \mu_c(\bar{D}) - \mu_G(\bar{D})$$

and similarly $\hat{\tilde{\mu}}_c(B) = \hat{\mu}_c(B) - \hat{\mu}_G(B)$. When clear, we may omit $\bar{D}$ and $B$ from the notation.

## C.2  LINEAR SEPARABILITY

In our study we are interested in quantifying the linear separability of the old tasks' classes in feature space. In this section we discuss the metric of linear separability used and derive a lower bound for it.

**Definition 2** (Linear Separability). Consider the two distributions $P_1$ and $P_2$. The *linear separability* of the two classes is defined as the maximum success rate achievable by any linear classifier:

$$\xi(P_1, P_2) := \max_{w, b} \left[ \mathcal{P}_{P_1}(w^\top x + b > 0) + \mathcal{P}_{P_2}(w^\top x + b < 0) \right].$$

Equivalently, $\xi(P_1, P_2) = 1 - \epsilon_{\min}$, where $\epsilon_{\min}$ is the minimal misclassification probability over all linear classifiers.

**Definition 3** (Mahalanobis Distance). Consider two distribution in the feature space $\mu_1, \mu_2$, and covariances $\Sigma_1, \Sigma_2$. The Mahalanobis distance between the two distributions is defined as

$$d_M^2(\mu_1, \mu_2, \Sigma_1, \Sigma_2) = (\mu_1 - \mu_2)^\top (\Sigma_1 + \Sigma_2)^{-1} (\mu_1 - \mu_2)$$

For two Gaussian distributions with equal covariance the Mahalanobis distance determines the minimal misclassification probability over all linear classifiers:

$$\epsilon_{min} = \Phi\left( -\tfrac{1}{2} \sqrt{d_M^2} \right)$$

In this study we take the Mahalanobis distance to be a proxy for the linear separability of two distributions in feature space. When only the first two moments of the distributions are known, this is the best proxy for linear separability. In the following lemma we derive a handy lower bound for the Mahalanobis distance which we will be using throughout.

**Lemma 1** (Lower Bound to Mahalanobis Distance). *Let $\mu_1, \mu_2 \in \mathbb{R}^d$ and $\Sigma_1, \Sigma_2 \in \mathbb{R}^{d \times d}$ be positive semidefinite covariance matrices. Then the squared Mahalanobis distance satisfies*

$$d_M^2(\mu_1, \mu_2, \Sigma_1, \Sigma_2) = (\mu_1 - \mu_2)^\top (\Sigma_1 + \Sigma_2)^{-1} (\mu_1 - \mu_2) \geq \frac{\|\mu_1 - \mu_2\|^2}{\operatorname{Tr}(\Sigma_1 + \Sigma_2)}.$$

*Proof.* Let $A := \Sigma_1 + \Sigma_2 \succeq 0$ and $v := \mu_1 - \mu_2$. Let $\lambda_i$ be the eigenvalues of $A$ and $u_i$ the corresponding orthonormal eigenvectors. Write

$$v = \sum_i \alpha_i u_i \quad \text{so that} \quad v^\top A^{-1} v = \sum_i \frac{\alpha_i^2}{\lambda_i}.$$

By Jensen's inequality for the convex function $f(x) = 1/x, x \in \mathbb{R}^+$ and the fact that $\sum_i \alpha_i^2 = \|v\|^2$, we have

$$\sum_i \frac{\alpha_i^2}{\lambda_i} \geq \frac{\sum_i \alpha_i^2}{\sum_i \lambda_i} = \frac{\|v\|^2}{\operatorname{Tr}(A)}.$$

Applying this to $v = \mu_1 - \mu_2$ and $A = \Sigma_1 + \Sigma_2$ gives the claimed inequality. $\square$

In this work we use the lower bound to the Mahalanobis distance as a proxy for linear separability. This quantity is also related to the *signal to noise ratio*, and thus hereafter we use the following

notation:

$$SNR(c_1, c_2) = \frac{\|\mu_1 - \mu_2\|^2}{\text{Tr}(\Sigma_1 + \Sigma_2)}$$

$SNR(c_1, c_2)$ and $\xi(c_1, c_2)$ are directly proportional, although the latter is bounded while the former is not. Therefore an increase in $SNR(c_1, c_2)$ corresponds to an increase in linear separability, within the applicability of a Gaussian assumption.

### C.3 TERMINAL PHASE OF TRAINING (TPT)

The *terminal phase of training* is the set of training steps including and succeeding the step where the training loss is zero. Given our network structure, a direct consequence of TPT is that the class-conditional distributions are *linearly separable* in feature space.

Starting from Papyan et al. (2020), several works have studied the structures that emerge in the network in this last phase of training (see Appendix B for an overview). In particular, Papyan et al. (2020) has discovered that TPT induces the phenomenon of *Neural Collapse* (NC) on the features of the training data. This phenomenon is composed of four key distinct effects, which we outline in the following definitions. Notably *the definitions below apply exclusively to the training data*, which we denote generically by $D$ here. Thus, the class means and the global means in Definition 5 are all computed using the training data (i.e. $\mu_c = \mu_c(D)$, and $\tilde{\mu}_c = \tilde{\mu}_c(D)$).

**Definition 4** (NC1 or Variability collapse). Let $t$ be the training step index and $\phi_t$ the feature map at step $t$ trained on data $D$. Then, the within-class variation becomes negligible as the features collapse to their class means. In other words, for every $x \in X_{D,c}$, with $c$ in the training data:

$$\mathbb{E}_{X_{D,c}}\left[\|\phi_t(x) - \mu_c(t)\|^2\right] = \delta_t, \qquad \lim_{t \to +\infty} \delta_t = 0 \tag{3}$$

**Definition 5** (NC2 or Convergence to Simplex ETF). The vectors of the class means (after centering by their global mean) converge to having equal length, forming equal-sized angles between any given pair, and being the maximally pairwise-distanced configuration constrained to the previous two properties.

$$\lim_{t \to +\infty} \|\tilde{\mu}_c(t)\|_2 \to \beta_t \ \forall c \tag{4}$$

$$\lim_{t \to +\infty} \cos(\tilde{\mu}_c(t), \tilde{\mu}_{c'}(t)) \to \begin{cases} 1 & \text{if } c = c' \\ -\frac{1}{K-1} & \text{if } c \neq c' \end{cases} \tag{5}$$

**Definition 6** (NC3 or Convergence to Self-duality). The class means and linear classifiers—although mathematically quite different objects, living in dual-vector spaces—converge to each other, up to rescaling. Let $\tilde{U}(t) = [\tilde{\mu}_1(t), \ldots, \tilde{\mu}_K(t)]$:

$$\frac{W_h^\top(t)}{\|W_h(t)\|} = \frac{\tilde{U}(t)}{\|\tilde{U}(t)\|} \tag{6}$$

As a consequence, $\text{rank}(W_h(t)) = \text{rank}(\tilde{U}(t)) = K - 1$.

**Definition 7** (NC4 or Simplification to NCC). For a given deepnet activation, the network classifier converges to choosing whichever class has the nearest train class mean (in standard Euclidean distance).

☞ **Notation .** In all the following proofs we denote by $S_t = \text{span}(\{\tilde{\mu}_1(t), \ldots, \tilde{\mu}_K(t)\})$ and by $S_t^\perp$ its orthogonal components, and similarly by $P_{S_t}, P_{S_t^\perp}$ the respective projection operators. Note that the reference to the training data is implicit. We might signal it explicitly when necessary.

**Lemma 2** (Feature classes gram matrix). *Let $\tilde{U}_t = [\tilde{\mu}_1(t), \ldots, \tilde{\mu}_K(t)]$ (computed with respect to the training data). Then there exist $t_0$ in the TPT such that, for all $t > t_0$ the gram matrix $\tilde{U}_t^\top \tilde{U}_t$ has the following structure:*

$$\tilde{U}_t^\top \tilde{U}_t = \beta \left( I_K - \tfrac{1}{K} \mathbf{1} \mathbf{1}^\top \right) \tag{7}$$

$$(\tilde{U}_t^\top \tilde{U}_t)^{-1} = \beta^{-1} \left( I_K - \tfrac{1}{2K} \mathbf{1} \mathbf{1}^\top \right) \tag{8}$$

*Proof.* Let $\tilde{\mu}_c(t) = \mu_c(t) - \mu_G(t)$ be the centered class mean given by $\phi_t$ on the training data. Then by Definition 5 we know that for all $t > t_0$ for some $t_0$:

$$\langle \tilde{\mu}_c(t), \tilde{\mu}_{c'}(t) \rangle = \begin{cases} \beta_t, & c = c', \\ -\frac{\beta_t}{K-1}, & c \neq c', \end{cases}$$

Also, denote by $\tilde{U}_t = [\tilde{\mu}_1(t), \ldots, \tilde{\mu}_K(t)]$ the matrix of centered class means. Then the centered Gram matrix $\tilde{U}_t^\top \tilde{U}_t$ has the following structure:

$$\tilde{U}_t^\top \tilde{U}_t = \beta_t \left( I_K - \tfrac{1}{K} \mathbf{1} \mathbf{1}^\top \right)$$

which is a rank-one perturbation of a diagonal matrix. In fact, the matrix is a projection matrix onto the space orthogonal to $\mathbf{1}$, scaled by $\beta_t$. It has eigenvalues $\beta_t$ with multiplicity $K - 1$ and $0$ with multiplicity $1$. Since it's a projection matrix, it is idempotent (up to the scaling factor $\beta_t$). Its inverse does not exist but the pseudo-inverse is well-defined.

$$\beta_t \left( I_K - \tfrac{1}{K} \mathbf{1} \mathbf{1}^\top \right)^{-1} = \frac{1}{\beta_t} \left( I_K - \tfrac{1}{K} \mathbf{1} \mathbf{1}^\top \right)$$

$\square$

### C.4 Neural Collapse in a continual learning setup

Depending on the continual learning setup, the number of outputs in the network may be increasing with each task. Therefore the Neural Collapse definitions need to be carefully revisited for different continual learning scenarios.

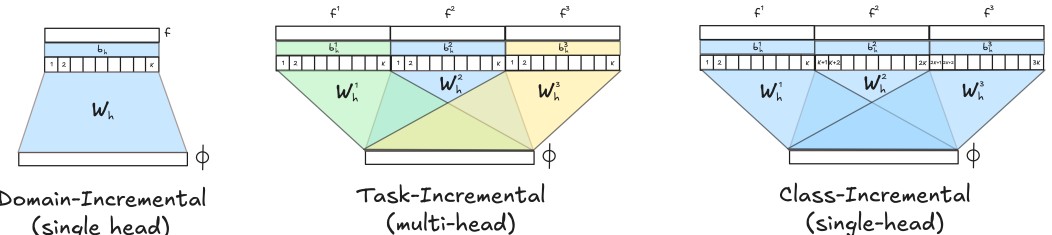

Figure 25: Depiction of continual learning Setups and corresponding head structures. Different colors indicate different gradient information propagated through the weights.

In the case of **task-incremental** and **class-incremental** learning, where each task introduces new classes, we distinguish between the tasks heads as follows:

$$f_\theta^{(i)}(x) = W_h^{(i)} \phi(x) + b_h^{(i)} \tag{9}$$

$$f_\theta(x) = [f_\theta^{(1)}(x), \ldots, f_\theta^{(i)}(x)]^\top \tag{10}$$

where only heads from the first to the current task are used in the computation of the network function. For brevity, hereafter we will denote by $W_h^A, b_h^A$ the concatenation of *active heads* at any task: for example, for task $n$, $W_h^A = [W_h^{(1)}, \ldots, W_h^{(n)}]$ and $f_\theta(x) = W_h^A \phi(x) + b_h^A$. In order to unify the notation, the same symbols will be used for domain-incremental learning where $W_h^A = W_h$ and $b_h^A = b_h$. The difference between task- and class-incremental learning is whether the residuals depend only on the current task output $f_\theta^{(n)}(x)$ or on the entire output $f_\theta(x)$, as we will explain shortly.

### C.4.1 DOMAIN-INCREMENTAL LEARNING (DIL)

In *Domain-Incremental Learning (DIL)*, the classification head is consistently shared across all tasks, as each task utilizes the same set of classes. Consequently, NC is expected to induce a fixed number of clusters in the feature space, corresponding to the total number of classes. Given that the same class appears in multiple tasks, we must distinguish between the *population mean* $\mu_c(\bar{D})$ and the *observed mean* $\hat{\mu}_c(\hat{D})$, where $\hat{D}$ is a generic training set. Generally, we expect NC properties (Definitions 4 to 7) to emerge on the training data $\hat{D}$. If the training data includes a buffer, all class means and the global mean will be computed including the buffer samples. Accounting for this, the NC characteristics emerge analogously to those in single-task training.

**Emergent Task-Wise Simplex Structure**    Curiously, our experiments also observe an emergent *within-task simplex structure*. When features are centered by the *task-wise feature means* (taking, for each task, all samples included in the buffer), we also observe the characteristic NC structure within the task. This finding is non-trivial, because the task-wise mean and the global mean are not the same. It seems, then, that during continual learning in DIL, Neural Collapse emerges on *two distinct levels simultaneously*. This dual emergence creates a highly constrained feature manifold, which substantially limits the degrees of freedom available for learning subsequent tasks. Our observations suggest that *a significantly more constrained version of NC emerges under the DIL paradigm compared to standard single-task training*.

### C.4.2 CLASS-INCREMENTAL LEARNING (CIL)

In Class-Incremental Learning (CIL), each task introduces *a new set of classes* (for simplicity, we assume the same number $K$ per task). For task $n$, the classification head is expanded by adding $W_h^{(n)}$ to form

$$W_h^A(t) = [W_h^{(1)}(t), \ldots, W_h^{(n)}(t)].$$

Nevertheless, training proceeds as in a single-task setting: residuals are shared across all outputs,

$$\frac{\partial \ell((x, y), f_\theta)}{\partial f_\theta} = \tilde{f}_\theta(x) - \tilde{y},$$

where both $\tilde{f}_\theta(x)$ and $\tilde{y}$ are vectors of dimension $n \times K$. For instance, $\tilde{f}_\theta(x) = f_\theta(x)$ for MSE loss, and $\tilde{f}_\theta(x) = \mathrm{softmax}(f_\theta(x))$ for cross-entropy loss, while $\tilde{y}$ corresponds to the one-hot encoding of $y$.

In CIL, *the composition and relative proportion* of classes in the training data affect the asymptotically optimal feature structure. If all classes are present in equal proportion, the Neural Collapse (NC) structure for task $n$ consists of $n \times K$ clusters with vanishing intra-cluster variance, which increases to $(n + 1) \times K$ clusters when the next task is introduced. By Definition 6, the resulting rank of the weight matrix is $n \times K - 1$ after $n$ tasks.

However, if the training dataset is imbalanced—i.e., the number of samples per class is not equal—the network is pushed, during the TPT, toward a variant of NC known as *Minority Collapse (MC)* (Fang et al., 2021). For this reason, in our experiments we use datasets with equal numbers of samples per class and buffers of uniform size across tasks. Assuming all tasks' datasets have the same size, for a dataset $D$ and buffer $B$, the degree of imbalance can be quantified by

$$\rho = \frac{|B|}{|D|}.$$

Dang et al. (2023); Hong & Ling (2023) identify a critical threshold for $\rho$: below this value, the heads of minority classes (i.e., buffer classes) become indistinguishable, producing nearly identical outputs for different classes. Above the threshold, the MC structure is gradually restored to a standard NC configuration, with class mean norms and angles increasing smoothly.

As noted by Fang et al. (2021), MC can be avoided by *over-sampling* from minority classes to restore class balance. In continual learning, this is implemented by sampling in a *task-balanced* fashion from the buffer, ensuring that each batch contains an equal number of samples per class. Under task-balanced sampling, the class-incremental setup reproduces the standard NC characteristics observed in single-task training. In contrast, in the absence of replay, class-incremental learning is inherently prone to Minority Collapse.

### C.4.3 TASK-INCREMENTAL LEARNING (TIL)

In Task-Incremental Learning (TIL), each task introduces $K$ new classes, as in the CIL case. The crucial difference lies in the treatment of the residuals: they are computed separately for each task. For a sample $x$ belonging to task $n$, we have

$$\frac{\partial \ell((x,y), f_\theta)}{\partial f_\theta} = \tilde{f}_\theta^{(n)}(x) - \tilde{y}^{(n)},$$

where both $\tilde{f}_\theta^{(n)}(x)$ and $\tilde{y}^{(n)}$ are $K$-dimensional vectors. For instance, under MSE loss $\tilde{f}_\theta^{(n)}(x) = f_\theta^{(n)}(x)$, while under cross-entropy $\tilde{f}_\theta^{(n)}(x) = \text{softmax}(f_\theta^{(n)}(x))$, and $\tilde{y}^{(n)}$ denotes the one-hot encoding of $y \in \{0, \dots, K-1\}$.

Since the outputs are partitioned across tasks, logits corresponding to inactive heads do not contribute to the loss. That is, for $x \in D_i$, the terms $W_h^{(j)}\phi(x) + b_h^{(j)}$ with $j \neq i$ remain unconstrained. In contrast, in CIL such logits are explicitly penalized, as the residuals are shared across all heads. Consequently, the TIL multi-head setting imposes fewer explicit constraints on the relative geometry of weights and class means across tasks.

Our empirical results indeed reveal that there is structure within each task, but the relative geometry across tasks is more variable and does not seem to exhibit a clear pattern. Within each task, the features exhibit the standard Neural Collapse (NC) geometry, consistent with Definitions 4 to 6. However, the class means of different tasks can overlap arbitrarily, as there are no explicit constraints linking them.

Motivated by these observations, we formalize the emergent structure as follows.

---

**Proposition 1** (Neural Collapse in Multi-Head Models). *Let $\mu_c^n(t)$ denote the mean feature of class $c$ from task $n$ at time $t$. In the terminal phase of training, under balanced sampling, the following hold:*

1. **NC1 (Variability collapse).** *Within each task, features collapse to their class means, i.e.,*

$$\lim_{t \to +\infty} \mathbb{E}_{x \in X_c^n} \left[ \|\phi_t(x) - \mu_c^n(t)\|^2 \right] = 0.$$

2. **NC2 (Convergence to simplex ETF within each task).** *Centered class means within each task converge to an Equiangular Tight Frame (ETF):*

$$\lim_{t \to +\infty} \|\tilde{\mu}_c^n(t)\|_2 \to \beta_t^n, \qquad \forall c \in \{1, \dots, K\}, \tag{11}$$

$$\lim_{t \to \infty} \cos(\tilde{\mu}_c^n(t), \tilde{\mu}_{c'}^n(t)) \to \begin{cases} 1 & \text{if } c = c', \\ -\frac{1}{K-1} & \text{if } c \neq c', \end{cases} \tag{12}$$

*where $\tilde{\mu}_c^n(t) = \mu_c^n(t) - \mu_G^n(t)$ and $\mu_G^n(t)$ is the task mean.*

3. **NC3 (Convergence to self-duality).** *The classifier weights for each head align with the centered class means of the corresponding task, up to rescaling:*

$$\frac{W_h^{(n)\top}(t)}{\|W_h^{(n)}(t)\|} = \frac{\tilde{U}^{(n)}(t)}{\|\tilde{U}^{(n)}(t)\|},$$

*where $\tilde{U}^{(n)}(t) = [\tilde{\mu}_1^n(t), \dots, \tilde{\mu}_K^n(t)]$. Consequently, $\text{rank}(W_h^{(n)}(t)) = \text{rank}(\tilde{U}^{(n)}(t)) = K - 1$.*

*In summary, each task forms an ETF simplex in the feature space (NC2), with variability collapse (NC1) and classifier self-duality (NC3) holding as in the single-task case.*

---

A key implication is the difference in rank scaling compared to CIL. In CIL, the rank of the head weights after $n$ tasks is $n \times K - 1$, whereas in TIL it is *upper bounded by* $n \times (K-1)$, as confirmed empirically (Figure 16). Thus, the multi-head structure imposes a strictly stronger rank limitation.

**Replay vs. no replay.** When training without replay, i.e., relying solely on the current task's data, the TIL setup reduces to an effective single-task regime: earlier heads receive no gradient signal, and NC emerges only within the most recent task, as in standard single-task training.

---

**Lemma 3** (Gram Matrix in TIL). *Let $\tilde{U}_t = [\tilde{U}_t^{(1)}, \ldots, \tilde{U}_t^{(n)}]$ be the matrix of centered class means at time t, where $\tilde{U}_t^{(m)} = [\tilde{\mu}_1^{(m)}(t), \ldots, \tilde{\mu}_K^{(m)}(t)]$, with $\tilde{\mu}_c^{(m)}(t) = \mu_c^{(m)}(t) - \mu_G^{(m)}(t)$. Suppose that in the terminal phase of training Proposition 1 holds. Then for all sufficiently large t, the Gram matrix $\tilde{U}_t^\top \tilde{U}_t$ is :*

$$\tilde{U}_t^\top \tilde{U}_t = \begin{bmatrix} G_t^{(1)} & \tilde{U}_t^{(1)\top} \tilde{U}_t^{(2)} & \cdots & \tilde{U}_t^{(n)\top} \tilde{U}_t^{(1)} \\ \tilde{U}_t^{(1)\top} \tilde{U}_t^{(2)} & G_t^{(2)} & \cdots & \tilde{U}_t^{(n)\top} \tilde{U}_t^{(2)} \\ \vdots & \vdots & \ddots & \vdots \\ \tilde{U}_t^{(n)\top} \tilde{U}_t^{(1)} & \tilde{U}_t^{(n)\top} \tilde{U}_t^{(2)} & \cdots & \tilde{G}_t^{(n)} \end{bmatrix},$$

*where each block $G_t^{(m)}$ satisfies*

$$G_t^{(m)} = \beta_m \left( I_K - \tfrac{1}{K} \mathbf{1}\mathbf{1}^\top \right), \qquad G_t^{(m)^{-1}} = \beta_m^{-1} \left( I_K - \tfrac{1}{K} \mathbf{1}\mathbf{1}^\top \right).$$

*Thus, the inverse Gram matrix satisfies $(\tilde{U}_t^\top \tilde{U}_t)^{-1} \mathbf{1} = \mathbf{0}$.*

---

*Proof.* By Proposition 1, in the terminal phase of training each task satisfies NC2 (within-task ETF) and task subspaces are orthogonal.

**Diagonal blocks:** Each $G_t^{(m)} = \tilde{U}_t^{(m)\top} \tilde{U}_t^{(m)}$ is an ETF matrix of size $K \times K$. By definition of ETF, its columns sum to zero:

$$G_t^{(m)} \mathbf{1} = 0.$$

**Off-diagonal blocks:** For $B^{(ij)} = \tilde{U}_t^{(i)\top} \tilde{U}_t^{(j)}$, we have

$$B^{(ij)} \mathbf{1} = \tilde{U}_t^{(i)\top} \tilde{U}_t^{(j)} \mathbf{1} = \tilde{U}_t^{(i)\top} \cdot 0 = 0$$

since the columns of $\tilde{U}_t^{(j)}$ are centered.

**Global null vector:** For the full block Gram matrix $\tilde{U}_t^\top \tilde{U}_t$, the $i$-th block-row acting on $\mathbf{1}_n$ is

$$\sum_{j=1}^n B^{(ij)} \mathbf{1} = G_t^{(i)} \mathbf{1} + \sum_{j \neq i} B^{(ij)} \mathbf{1} = 0 + \sum_{j \neq i} 0 = 0.$$

Hence, $\tilde{U}_t^\top \tilde{U}_t \mathbf{1}_n = 0$, so $\mathbf{1}_n$ lies in the null space of the Gram matrix.

**Inverse / pseudoinverse:** Since $\tilde{U}_t^\top \tilde{U}_t$ is singular, the Moore–Penrose pseudoinverse exists, and $\mathbf{1} \in \ker(\tilde{U}_t^\top \tilde{U}_t)$ implies $\mathbf{1} \in \ker((\tilde{U}_t^\top \tilde{U}_t)^+)$. Thus, $\mathbf{1}$ is a zero eigenvector of both $\tilde{U}_t^\top \tilde{U}_t$ and its pseudoinverse. □

### C.4.4 FINAL RESULTS AND TAKEAWAYS

The preceding analysis allows us to draw several unifying conclusions regarding the asymptotic feature geometry in continual learning.

A first key takeaway is that, in the absence of replay, continual learning effectively reduces to repeated single-task training. In this regime, only the current task is represented in feature space with Neural Collapse (NC) geometry, while features from previous tasks degenerate. This observation is formalized as follows.

---

**Finding 1** (Asymptotic Structure without replay). *When training exclusively on the most recent task, irrespective of the continual learning setup, the asymptotically optimal feature representation for the current task coincides with the Neural Collapse (NC) structure observed in the single-task regime. In the CIL case, this further implies that the feature representations of all classes from previous tasks collapse to the zero vector, while only the features of the current task organize according to NC.*

---

A second key takeaway is that task-balanced replay fundamentally alters the asymptotic structure. In this setting, the replay buffer restores balanced exposure to all classes, preventing the degeneration of past representations. Consequently, in single-head setups (DIL and CIL) the network converges to a global NC structure over all observed classes (measured on the training data). In contrast, the multi-head setup of TIL continues to decouple the heads across tasks, yielding NC geometry within each task but leaving the relative geometry across tasks unconstrained.

> **Finding 2** (Asymptotic Structure of the Feature Space with task-balanced replay). *When training on $n$ tasks with task-balanced replay, the single-head setups converge to Neural Collapse over all classes represented in the training data ($K$ classes for DIL and $n \times K$ classes for CIL). For TIL, each task head individually exhibits Neural Collapse within its $K$ classes, but the relative positioning of class means across tasks is unconstrained, leading to a blockwise NC structure in feature space.*

Taken together, these results highlight a fundamental distinction between single-head and multi-head continual learning: while replay suffices to recover global NC geometry in single-head settings, in TIL the absence of cross-task coupling in the loss function enforces only local NC structure within each task.

## C.5   MAIN RESULT 1: STABILIZATION OF THE TRAINING FEATURE SUBSPACE.

> **Theorem 4** (Subspace stabilization in TPT under SGD.). *Let $f_{\theta_t}(x) = W_h^A(t)\,\phi_t(x) + b_h^A(t)$ be the network at step $t$ in the optimization of a task with $P$ classes and dataset $D$, and let $S_t = \text{span}(\{\tilde{\mu}_1(t), \dots, \tilde{\mu}_P(t)\})$ $(\mu_c = \mu_c(D))$. Assume NC3 holds on $D$ for all $t \geq t_0$, i.e., $\text{span}(W_h^A(t)) = S_t$. Then, for all $t \geq t_0$, the gradient $\nabla_\theta \mathcal{L}(\theta_t)$ is confined to directions in parameter space that affect features in $S_t$, and, consequently, $S_t = S_{t_0}$ and $S_t^\perp = S_{t_0}^\perp$.*

*Proof.* Let $\phi_t(x)$ be the feature representation of $x$ at time $t$, and let $J_t(x) = \nabla_\theta \phi_t(x)$ be its Jacobian with respect to parameters $\theta_t$. Consider an infinitesimal parameter change $\Delta\theta_t = \epsilon\,v$, with $P_{S_t} J_t(x) v = 0$ for all $x$ in the training data, i.e., this change only affects the feature component in $S_t^\perp$. By a first order approximation the corresponding feature change is:

$$\Delta\phi_t(x) = J_t(x)\Delta\theta_t = \epsilon\,J_t(x)v = \epsilon\,P_{S_t^\perp} J_t(x)v$$

Now, consider the effect of this change on the loss:

$$\mathcal{L}(\theta_t + \epsilon\,v) - \mathcal{L}(\theta_t) \approx \nabla_\phi \mathcal{L}(\theta_t) \cdot \Delta\phi_t(x) \tag{13}$$

$$= \left(\frac{\partial \mathcal{L}}{\partial f} \cdot \frac{\partial f}{\partial \phi}\right) \cdot \Delta\phi_t(x) \tag{14}$$

$$= \left(\frac{\partial \mathcal{L}}{\partial f} \cdot W_h^A(t)\right) \cdot \Delta\phi_t(x) \tag{15}$$

By NC3, for any $t > t_0$, $\text{span}(W_h^A(t)) = S_t$, and since $\Delta\phi_t(x) \in S_t^\perp$:

$$W_h^A(t) \cdot \Delta\phi_t(x) = 0 \Rightarrow \mathcal{L}(\theta_t + \epsilon\,v) - \mathcal{L}(\theta_t) = 0$$

Dividing by $\epsilon$ and taking the limit $\epsilon \to 0$,

$$\nabla_\theta \mathcal{L}(\theta_t) \perp v \quad \text{for all } v \text{ such that } P_{S_t} J_t(x)v = 0 \; \forall\, x \in D$$

This shows that the loss gradient lies entirely in directions that affect $S_t$ and consequently *the $S_t^\perp$ component of the input representation is not changed.* It follows that, after NC3 gradient descent cannot change the subspaces $S_t, S_t^\perp$, since all changes in the features for $t > t_0$ will lie in $S_{t_0}$. We conclude that $S_t = S_{t_0}$ and $S_t^\perp = S_{t_0}^\perp$. $\qquad\square$

☞ **Notation .** Hereafter we denote by $S$ the subspace spanned by the centered class means after its stabilization at the onset of NC3, i.e. $S = S_{t_0}$. Note that the centered class means may still change, but their span doesn't.

**Lemma 4** (Freezing and decay of $S^\perp$ in TPT under SGD.). *Let $f_{\theta_t}(x) = W_h^A(t)\phi_t(x) + b_h^A(t)$ be the network at time $t$, where $\phi_t(x)$ is the feature representation and $W_h^A(t)$ the final layer weights. Suppose the training loss includes weight decay with coefficient $\lambda > 0$, i.e.,*

$$\mathcal{L}_{total}(\theta) = \mathcal{L}(\theta) + \frac{\lambda}{2}\|\theta\|^2.$$

*and that for all $t \geq t_0$, NC3 holds, i.e., $\mathrm{span}(W_h^A(t)) = S$, and $\eta$ sufficiently small. Then the component of $\phi_t(x)$ in $S^\perp$, denoted by $\phi_{t,S^\perp}(x)$, evolves as follows:*

$$\phi_{t,S^\perp}(x) = \upsilon^{t-t_0}\,\phi_{t_0,S^\perp}(x)$$

*Proof.* By gradient descent the parameter update is:

$$\Delta\theta_t = -\eta\left(\nabla_\theta\mathcal{L}(\theta_t) + \lambda\theta_t\right)$$

and, for small enough $\eta$ we can approximate the feature update as :

$$\phi_{t+1}(x) - \phi_t(x) \approx J_t(x)\,\Delta\theta_t = -\eta\,J_t(x)\,\nabla_\theta\mathcal{L}(\theta_t) - \eta\lambda\,J_t(x)\,\theta_t$$

Decompose this into components in $S$ and $S^\perp$. By Theorem 4, for all $t > t_0$ and all $x \in D$ $J_t(x)\,\nabla_\theta\mathcal{L}(\theta) \in S$. Then:

$$\phi_{t+1,S^\perp}(x) - \phi_{t,S^\perp}(x) = -\eta\lambda P_{S^\perp}J_t(x)\theta_t$$

Noticing that $\theta = 0$ makes $\phi(x) = 0$ for any $x$, by a first order approximation we have that $\phi_t(x) \approx J_t(x)\,\theta_t$ and thus:

$$\phi_{t+1,S^\perp}(x) = \phi_{t,S^\perp}(x)(1 - \eta\lambda)$$

for all $t > t_0$. Unrolling this sequence over time, starting from $t_0$, we get our result. $\square$

*Remark.* The results presented in this section hold for both single-head and multi-head training. When training with more than 1 head, the subspace $S$ corresponds to the span of the class means of all heads combined, and by Proposition 1 it has lower rank than in the single-head case.

C.6 ANOTHER DEFINITION OF OOD

**Definition 8** (ID/OOD orthogonality property of Ammar et al. (2024)). Consider a model with feature map $\phi_t(x)$, trained on dataset $D$ with $K$ classes. Denote by $S_t = \mathrm{span}\{\tilde{\mu}_1(t),\ldots,\tilde{\mu}_K(t)\}$ the subspace spanned by the centered class means of the training data at time $t$. The set of data $X$ is said to be OOD if

$$\cos\left(\mathbb{E}_X[\phi_t(x)], \mu_c(t)\right) \to 0 \qquad \forall\, c \in [K]$$

**Definition 9** (Out-of-distribution (OOD)). Let $X_c$ be a set of samples from class $c$. Consider a network with feature map, $\phi_t(x)$, trained on dataset $D$ with $K$ classes, such that $X_c \cap D = \emptyset$. Denote by $S_t = \mathrm{span}\{\tilde{\mu}_1(t),\ldots,\tilde{\mu}_K(t)\}$ the subspace spanned by the centered class means of the training data at time $t$. We say that $X_c$ is *out of distribution* for $f_{\theta_t}$ (trained on $D$) if

$$P_{S_t}\mathbb{E}_{X_c}[\phi(x)] = 0$$

This definition restates the *ID/OOD orthogonality* property of Ammar et al. (2024) in a different form.

Next, we show that the observation, common in the OOD detection literature, that old tasks data is maximally uncertain in the network output is coherent with these definitions of OOD when there is Neural Collapse.

**Proposition 2** (Out Of Distribution (OOD) data is maximally uncertain.). *A set of samples $X$ from the same class $c$ is* out of distribution *for the model $f_\theta$ with homogeneous head and Neural Collapse if and only if the average model output over $X$ is maximally uncertain, i.e. the uniform distribution.*

*Proof.* By definition of $S$ being the span of $\{\tilde{\mu}_1(t), \ldots, \tilde{\mu}_K(t)\}$ we can write

$$P_{\tilde{U}}(t)\,\phi_t(x) = \tilde{U}(t)(\tilde{U}(t)^\top \tilde{U}(t))^{-1}\tilde{U}(t)^\top \phi_t(x)$$

where $\tilde{U}$ is the matrix whose columns are the centered class means $\tilde{\mu}_i$. By Definition 6 we have, for all $t > t_0$, $W_h(t) = \alpha\,\tilde{U}_t$, where $\alpha = \frac{\|W_h(t)\|}{\|\tilde{U}(t)\|}$ and therefore, for an homogenous head model, the network outputs are $f_\theta(x) = \tilde{U}(t)^\top \phi_t(x)$ . Finally, to complete the proof see that by the structure of the gram matrix (Lemma 2), its null space is one-dimensional along the $\mathbf{1}$ direction. Therefore it must be that

$$\tilde{U}(t)^\top \mathbb{E}_{X_c}[\phi(x)] \propto \mathbf{1} \tag{16}$$

$$P_{\tilde{U}}(t)\,\mathbb{E}_{X_c}[\phi(x)] = 0 \tag{17}$$

are always true concurrently. $\qquad\square$

*Remark* (Old task data behaves as OOD without replay). When training on task $n$ without replay, samples from previous tasks $m < n$ effectively behave as out-of-distribution for the *active subspace* corresponding to task $n$, in the sense of Definition 9. For single-head models, a similar effect occurs in CIL due to *Minority Collapse*, which guarantees that the representations $\phi_t(x)$ of old task data simply converges to the origin, which is trivially orthogonal to $S_t$. Consequently, the theoretical results we derive for OOD data in this section also apply to old task data under training without replay.

**Corollary 3** (The OOD class mean vector converges to 0 in TPT under SGD with weight decay.)**.**
*In the TPT, with weight decay coefficient $\lambda > 0$, OOD class inputs $X_c$ are all mapped to the origin asymptotically*

$$\lim_{t\to\infty} \mathbb{E}_{X_c}[\phi_t(x)] = 0$$

## C.7 Asymptotics of OOD data

☞ **Notation .** To simplify exposition, we introduce the notation $\upsilon = 1 - \eta\lambda$. Additionally, in this section we use $W_h$ and $\tilde{U}$ to refer in general to the head and class means used in the current training. Note that, since we don't consider replay for now, this is equivalent to the current task's classes' head and features.

**Theorem 5** (OOD class variance after NC3.)**.** *Let $b_t(x)$ be the coefficients of the projection of the input $x$ on the centered training class means space $S$. In the terminal phase of training, for OOD inputs, if $b_t(x)$, $x \in X_c$ has covariance $\Sigma_c$ with constant norm in $t$, then the within-class variance in feature space for $X_c$ satisfies*

$$Var_{X_c}(\phi_t(x)) \in \Theta\Big(\beta_t^A + (1 - \eta\lambda)^{2(t-t_0)}\Big), \tag{18}$$

*where $\beta_t^A$ accounts for the contribution of all active heads: in the single-head case $\beta_t^A = \beta_t$, and in the multi-head case $\beta_t^A = \sum_{m=1}^n \beta^m$ with $n$ the number of active heads.*

*Proof.* Consider representations of inputs from an OOD class $X_c$. By Theorem 4, for any $t \geq t_0$, we can decompose

$$\phi_t(x) = \phi_{t,S}(x) + \phi_{t_0,S^\perp}(x),$$

where $\phi_{t,S}(x)$ lies in the span of the centered training class means. We can express this component as

$$\phi_{t,S}(x) = \tilde{U}(t)\,b_t(x), \quad b_t(x) = (\tilde{U}(t)^\top \tilde{U}(t))^{-1}\tilde{U}(t)^\top \phi_t(x).$$

From Definition 9, $\mathbb{E}_{X_c}[\phi_{t,S}(x)] = 0$. Hence, the within-class variance in feature space is

$$\mathrm{Var}_{X_c}(\phi_t(x)) = \mathbb{E}_{X_c}[\|\phi_t(x) - \mathbb{E}_{X_c}[\phi_{t,S^\perp}(x)]\|^2] \tag{19}$$

$$= \mathbb{E}_{X_c}[\|\tilde{U}(t)\,b_t(x)\|^2] + \mathrm{Var}_{X_c,S^\perp}(\phi_t(x)). \tag{20}$$

The orthogonal component $S^\perp$ shrinks or remains constant due to Lemma 4:

$$\mathrm{Var}_{X_c, S^\perp}(\phi_t(x)) = (1 - \eta\lambda)^{2(t-t_0)} \mathrm{Var}_{X_c, S^\perp}(\phi_{t_0}(x)).$$

The variance in the $S$ component depends on the covariance $\Sigma_c$ of $b_t(x)$, which is assumed constant in $t$:

$$\mathrm{Cov}_{X_c}[\phi_{t,S}(x)] = \tilde{U}(t)\Sigma_c\tilde{U}(t)^\top.$$

Thus,

$$\mathrm{Var}_{X_c, S}(\phi_t(x)) = \mathrm{tr}(\tilde{U}(t)\Sigma_c\tilde{U}(t)^\top) = \mathrm{tr}(A\,\Sigma_c),$$

where $A = \tilde{U}(t)^\top\tilde{U}(t)$ has the structure described in Definition 5 and Proposition 1.

**Single-head case.** For $P$ classes, $A$ is an ETF matrix with $P$ vertices

$$A_{kk} = \beta_t, \quad A_{jk} = -\frac{\beta_t}{P-1}, \quad j \neq k,$$

so that

$$\beta_t \underbrace{\frac{P}{P-1}\big(\mathrm{tr}(\Sigma_c) - \lambda_1(\Sigma_c)\big)}_{C_{\mathrm{low}}} \leq \mathrm{tr}(A\,\Sigma_c) \leq \beta_t \underbrace{\frac{P}{P-1}\mathrm{tr}(\Sigma_c)}_{C_{\mathrm{high}}}.$$

**Multi-head case.** For $n$ heads, $A$ has the block structure described in Lemma 3, with each diagonal block having $K-1$ eigenvalues equal to $\beta^m$ and one zero eigenvalue. Hence,

$$\sum_{m=1}^{n} \beta_t^m \frac{K}{K-1}\underbrace{\big(\mathrm{tr}(\Sigma_c^{(m)}) - \lambda_1(\Sigma_c^{(m)})\big)}_{\geq C_{\mathrm{low}}} \leq \mathrm{tr}(A\,\Sigma_c) \leq \sum_{m=1}^{n} \beta_t^m \frac{K}{K-1}\underbrace{\mathrm{tr}(\Sigma_c^{(m)})}_{\leq C_{\mathrm{high}}}.$$

Denoting by $\beta_t^A = \frac{1}{n}\sum_1^n \beta_t^m$ we get:

$$\beta_t^A \frac{P}{K-1} C_{\mathrm{low}} \leq \mathrm{tr}(A\,\Sigma_c) \leq \beta_t^A \frac{P}{K-1} C_{\mathrm{high}}.$$

Thus, recognising that the only dynamic variable in $t$ is $\beta_t^A$ for both cases, we obtain

$$\mathrm{Var}_{X_c, S}(\phi_t(x)) \in \Theta(\beta_t^A), \quad \mathrm{Var}_{X_c, S^\perp}(\phi_t(x)) \in \Theta\big((1-\eta\lambda)^{2(t-t_0)}\big),$$

completing the proof. $\qquad\square$

☞ **Notation .** When we are not considering replay, there is only one active head in multi-headed models. In this cases we use $\beta_t$ to denote the feature norm of the active head. The results of this section are presented in a more general way, using $\beta_t^A$ to denote the contribution of all active heads.

**Theorem 6** (Linear separability of OOD data with Neural Collapse.). *Consider two OOD classes with inputs $X_{c_1}, X_{c_2}$. During TPT of the model $f_{\theta_t}(x)$ trained on a dataset $D$, the SNR between the two classes has asymptotic behaviour:*

$$SNR(c_1, c_2) \in \Theta\left(\left(\frac{\beta_t^A}{(1-\eta\lambda)^{2(t-t_0)}} + 1\right)^{-1}\right)$$

*where $\beta_t^A$ is the class feature norm, averaged across the active heads.*

*Proof.* Let $P_{X_{c_1}}(\phi_t(x))$, $P_{X_{c_2}}(\phi_t(x))$ be the distributions of the two OOD classes in feature space. Let $\mu_1, \mu_2$ and $\Sigma_1, \Sigma_2$ be the respective mean and covariances in feature space. By Definition 9 we know that $\mu_i = \mathbb{E}_{X_{c_i}}[\phi_{t,S^\perp}(x)]$ $(i = 1, 2)$. Therefore the SNR lower bound is:

$$SNR(c_1, c_2) = \frac{\|\mathbb{E}_{X_{c_1}}[\phi_{t,S^\perp}(x)] - \mathbb{E}_{X_{c_2}}[\phi_{t,S^\perp}(x)]\|^2}{\mathrm{Tr}(\Sigma_1 + \Sigma_2)}$$

where $\|\mathbb{E}_{X_{c_1}}[\phi_{t,S^\perp}(x)] - \mathbb{E}_{X_{c_2}}[\phi_{t,S^\perp}(x)]\|^2 \in \Theta\left((1-\eta\lambda)^{2(t-t_0)}\right)$. Notice that the trace decomposes across subspaces as well and therefore:

$$\mathrm{Tr}(\Sigma_1 + \Sigma_2) = \mathrm{Tr}(\Sigma_{1,S} + \Sigma_{1,S^\perp} + \Sigma_{2,S} + \Sigma_{2,S^\perp})$$

In the proof of Theorem 5 we have that $\mathrm{Tr}(\Sigma_{i,S}) \in \Theta(\beta)$ and $\mathrm{Tr}(\Sigma_{i,S^\perp}) \in \Theta\left((1-\eta\lambda)^{2(t-t_0)}\right)$. Thus from a simple asymptotic analysis we get that the linear separability of OOD data grows as:

$$SNR(c_1, c_2) \in \Theta\left(\left(\frac{\beta_t^A}{(1-\eta\lambda)^{2(t-t_0)}} + 1\right)^{-1}\right)$$

$\square$

*Remark.* By Theorem 6, when learning a new task without replay, if a class from a previous task becomes out-of-distribution (OOD) with respect to the current network (and its active subspace), an increasing class means norm $\beta_t$ or weight decay leads to *deep forgetting*, with the class information to degrade over time.

*Remark.* The SNR also depends on the degree of linear separability of the classes in the orthogonal subspace $S^\perp$ at the onset of NC. Consequently, in the absence of weight decay or without growth of the feature norms, the old classes may retain a nonzero level of linear separability asymptotically.

## C.8 MAIN RESULTS 3: FEATURE SPACE ASYMPTOTIC STRUCTURE WITH REPLAY.

We now turn our attention to training with replay, to explain how replay mitigates deep forgetting.

☞ **Notation.** We denote by $D_i$ the datasets of task $i$ and by $B_i$ the buffer used when training on task $n > i$. Further, let $\rho_i = |B_i|/|D_i|$ be the percentage of the dataset used for replay and assume that there is *balanced sampling*, i.e. each task is equally represented in each training batch. We again look at the case where there is Neural Collapse on the training data in TPT, which in this case is the current task data $D_n$ and the buffers $B_1, \ldots, B_{n-1}$. Finally, for DIL we denote by $X_c^i$ the data of class $c$ in task $i$ and by $X_c$ the data of class $c$ in all tasks, i.e. $X_c = \cup_{i=1}^n X_c^i$.

**Modeling the distribution of data from old tasks with replay**    Hereafter, we denote by $\hat{\mu} := \mu(B)$, the mean computed on the buffer samples. Define

$$\hat{\mu}_c(t) = \mu_c(t) + \xi_c(t)$$

where $\xi_c(t)$ is the difference between the *population mean* and the *observed mean*. For CIL and TIL this is the buffer $B_c$, while for DIL this is the union of all buffers $B = \cup_{i=1}^{n-1} B_i$ and the current task class data $X_c^n$. We know $\|\hat{\mu}_c(t) - \mu_c(t)\|$ decreases with the buffer size $b$ and, in particular, it's zero when $B_c = X_c$.

Let $\mathcal{D}_{NC}$ be the distribution of the representations when training on $100\%$ of the training data $X_c$. We know that this distribution has NC, each class $c$ has mean $\mu_c$ and decaying variance $\delta_t$. Also let $\mathcal{D}_{OOD}$ denote the OOD data distribution which we observe in the absence of replay (mean in $S^\perp$ and larger variance governed by $\beta_t$ and the decay factor $v^{t-t_0}$). Based on these observations, we model the distribution of $\phi_t(x)$ as the mixture of its two limiting distributions with mixing weight $\pi_c(b) \in [0,1]$ which is a monotonic function of $b$:

$$\phi_t(x) \sim \pi_c \, \mathcal{D}_{NC} + (1 - \pi_c) \, \mathcal{D}_{OOD}$$

According to this model, the mean and variance for the distribution of class $c$ asymptotically are:

$$\mu_c(t) = \pi_c \left(\hat{\mu}_c(t) + \xi_{c,S}(t)\right) + (1 - \pi_c) \left(v^{t-t_0} \, \mu_{c,S^\perp}(t_0)\right) \tag{21}$$

$$\sigma_c^2(t) = \Theta\left(\pi_c^2 \delta_t + (1 - \pi_c)^2 \left(\beta_t^A + v^{2(t-t_0)}\right)\right) \tag{22}$$

Note that in Equation (21) $S$ is defined based on $\pi_c$, and we absorbed the $S^\perp$ component of $\xi_c(t)$ in $\mu_{c,S^\perp}(t_0)$. In the variance expression we used the results of Theorem 5 for the OOD component and the fact that the variance of $\mathcal{D}_{NC}$ is $\delta_t$. In the TIL case, $\beta_t^A$ is the average of the class feature means across all active heads.

*Remark* (Interpretation of the buffer–OOD mixture model). The proposed model interpolates between two limiting regimes smoothly, and is based on our hypothesis regarding the evolution of the feature representation of past tasks as the buffer size is gradually increased. For small buffer size $b$, the representation distribution is dominated by the OOD component $\mathcal{D}_{OOD}$, which contributes variance in the orthogonal subspace $S^\perp$ and acts as structured noise with respect to the span $S$ of the current task. As $b$ increases, the mixture weight $\pi_c$ grows monotonically, and the replayed samples increasingly constrain the class means inside $S$. In the limit $b = |X_c|$, $\pi_c = 1$ and the representation collapses to the Neural Collapse distribution $\mathcal{D}_{NC}$ with vanishing variance. For intermediate $b$, the replay buffer introduces signal in $S$ through the term $\hat{\mu}_c(t)$, while the residual OOD component adds noise. The evolution of $\pi_c$ therefore captures how replay gradually aligns the buffer distribution with the NC structure, while modulating the relative strength of signal (from in-span replay) versus noise (from OOD drift).

**Proposition 3** (Concentration of buffer estimates). *Let $\mathcal{D}_c$ be the feature distribution of class $c$ at time $t$, with mean $\mu_{c,S}(t)$ in the active subspace $S$ and covariance $\Sigma_c$. Let $B_c \subset \mathcal{D}_c$ denote a replay buffer of size $b$ obtained by i.i.d. sampling. Then the buffer statistics $\hat{\mu}_c$ and $\hat{\Sigma}_c$ satisfy*

$$\mathbb{E}\big[\|\hat{\mu}_c - \mu_c\|^2\big] = O\left(\frac{\mathrm{Tr}(\Sigma)}{b}\right), \qquad \mathbb{E}\big[\|\hat{\Sigma}_c - \Sigma_c\|_F^2\big] = O\left(\frac{\mathrm{Tr}(\Sigma)}{b}\right).$$

*In particular, the standard deviation of both estimators decays as $O(b^{-1/2})$.*

*Proof.* Let $\{x_i\}_{i=1}^b \sim \mathcal{D}_c$ be i.i.d. samples with mean $\mu = \mu_c$ and covariance $\Sigma = \Sigma_c$. The sample mean satisfies $\hat{\mu}_c - \mu = \frac{1}{b}\sum_{i=1}^b (\phi(x_i) - \mu)$, so by independence ((Vershynin, 2018)),

$$\mathbb{E}\big[\|\hat{\mu}_c - \mu\|^2\big] = \frac{1}{b^2}\sum_{i=1}^b \mathbb{E}\big[\|\phi(x_i) - \mu\|^2\big] = \frac{1}{b}\mathrm{Tr}(\Sigma) = O\left(\frac{\mathrm{Tr}(\Sigma)}{b}\right).$$

Similarly, for the buffer covariance $\hat{\Sigma}_c$ we have

$$\mathbb{E}\big[\|\hat{\Sigma}_c - \Sigma\|_F^2\big] = O\left(\frac{\mathrm{Tr}(\Sigma)}{b}\right),$$

Thus the standard deviations of both estimators decay as $O(b^{-1/2})$. $\qquad\square$

In the above result we have hidden many other constants as they are independent of training time.

**Remark.** This bound should be interpreted as a heuristic scaling law rather than a formal guarantee. The key caveat is that feature evolution $\phi_t(x)$ is coupled to the buffer $B_c$ through training, violating independence. Nevertheless, the i.i.d. assumption is reasonable if buffer-induced correlations are small relative to the intrinsic variance of the features. In this sense, the bound captures the typical order of fluctuations in $\xi_c(t)$, even if the exact constants may differ in practice.

**Theorem 7** (Linear separability of replay data under Neural Collapse). *Let $c_1, c_2$ be two replay-buffer classes decoded by the same head, and let $\hat{\mu}_i(t)$ denote their observed class means with deviation $\xi_{i,S}(t)$ from the population mean inside the NC subspace $S$. Assume that the old classes features follow the mixture model*

$$\phi_t(x) \sim \pi_i\,\mathcal{D}_{NC} + (1 - \pi_i)\,\mathcal{D}_{OOD},$$

*with mixing proportion $\pi_i$, and that the class means norms for each task $m$ follow the same growth pattern $\beta_t^m \in \Theta(\beta_t)$. Then the signal-to-noise ratio between $c_1$ and $c_2$ satisfies*

$$SNR(c_1, c_2) \in \Theta\left(\frac{r^2\,\beta_t^A + v^{2(t-t_0)}}{r^2\,\delta_t + (\beta_t^A + v^{2(t-t_0)})}\right), \qquad r^2 = \frac{(\pi_1+\pi_2)^2}{(1-(\pi_1+\pi_2))^2}$$

*Proof.* Let $\mu_i(t), \Sigma_i(t)$ be the mean and covariance of class $i$ in feature space. If there is replay we assume they follow the mixed distribution described above with mixing proportion $\pi_1, \pi_2$ respec-

tively. Therefore, for each of them we know the following:

$$\mu_i(t) = \pi_i \left( \hat{\mu}_i(t) + \xi_{i,S}(t) \right) + (1 - \pi_i) \left( \upsilon^{t-t_0} \, \mu_{i,S^\perp}(t_0) \right) \tag{23}$$

$$\Sigma_i(t) = \pi_i^2 \, \Sigma_i^{NC}(t) + (1 - \pi_i)^2 \, \Sigma_i^{OOD}(t) \tag{24}$$

Moreover, by Theorem 5 we know that

$$\mathrm{tr}\left( \Sigma_i^{OOD}(t) \right) \in \Theta \left( \beta_t^A + \upsilon^{2(t-t_0)} \right)$$

and by Definition 4 we also know that $\mathrm{tr}\left( \Sigma_i^{NC}(t) \right) = \delta_t \to 0$ with $t \to +\infty$. Using this, we can write the SNR lower bound:

$$SNR(c_1, c_2) = \frac{\|\mu_{1,S}(t) - \mu_{2,S}(t)\|^2 + \|\mu_{1,S^\perp}(t) - \mu_{2,S^\perp}(t)\|^2}{\mathrm{Tr}(\Sigma_1(t) + \Sigma_2(t))}$$

where by definition of $\mu_i(t)$:

$$\|\mu_{1,S}(t) - \mu_{2,S}(t)\|^2 = \|\pi_1 \hat{\mu}_{1,S}(t) - \pi_2 \hat{\mu}_{2,S}(t) + \pi_1 \xi_{1,S} - \pi_2 \xi_{2,S}\|^2$$

$$\|\mu_{1,S^\perp}(t) - \mu_{2,S^\perp}(t)\|^2 = (\pi_1 - \pi_2)^2 \|\mu_G\|^2 + \upsilon^{t-t_0} \|(1 - \pi_1) \, \mu_{1,S^\perp}(t_0) - (1 - \pi_2) \, \mu_{2,S^\perp}(t_0)\|^2$$

Using the linearity of the trace and the fact that it decomposes across subspaces:

$$\mathrm{Tr}(\Sigma_i(t)) = \pi_i^2 \, \mathrm{Tr}(\Sigma_i^{NC}(t)) + (1 - \pi_i)^2 \, \mathrm{Tr}(\Sigma_i^{OOD}(t)) \in \Theta \left( \pi_i^2 \, \delta_t + (1 - \pi_i)^2 \left( \beta_t^A + \upsilon^{2(t-t_0)} \right) \right)$$

The mean difference in the $S$ component expands into

$$\|\pi_1 \hat{\mu}_{1,S} - \pi_2 \hat{\mu}_{2,S}\|^2 + \|\pi_1 \xi_{1,S} - \pi_2 \xi_{2,S}\|^2 - 2\langle \pi_1 \tilde{\mu}_1(B) - \pi_2 \tilde{\mu}_2(B), \, \pi_1 \xi_{1,S} - \pi_2 \xi_{2,S} \rangle$$

and the first term

$$\|\pi_1 \tilde{\mu}_1(B) - \pi_2 \tilde{\mu}_2(B)\|^2 = \pi_1^2 \|\tilde{\mu}_1(B)\|^2 + \pi_2^2 \|\tilde{\mu}_2(B)\|^2 + 2\pi_1 \pi_2 \langle \tilde{\mu}_1(B), \tilde{\mu}_2(B) \rangle$$

By Definition 5 and Proposition 1, and by the fact that $c_1, c_2$ belong to the same head $m$, also in multi-headed models, we know that $\|\tilde{\mu}_1(B)\|^2 = \|\tilde{\mu}_2(B)\|^2 \approx \beta_t$ and $\langle \mu_{c_1}, \mu_{c_2} \rangle = -\frac{\beta_t^A}{K-1}$. Then:

$$\|\pi_1 \tilde{\mu}_1(B) - \pi_2 \tilde{\mu}_2(B)\|^2 = (\pi_1^2 + \pi_2^2)\beta_t + 2\pi_1 \pi_2 \frac{\beta_t^A}{K-1} \in \Theta((\pi_1 + \pi_2)^2 \beta_t^A) \tag{25}$$

Define the per-class ratios

$$\eta_1 := \frac{\|\xi_{1,S}\|}{\|\tilde{\mu}_1(B)\|}, \qquad \eta_2 := \frac{\|\xi_{2,S}\|}{\|\tilde{\mu}_2(B)\|}.$$

Notice that the deviations in $S$ must behave in norm as the variance in the $S$ component, which by Proposition 3, $\mathrm{Tr}(\Sigma_i(t)) \in \Theta \left( \beta_t^A \right)$. Thus the coefficients satisfy $\eta_1, \eta_2 = \Theta(1)$. By the Cauchy-Schwarz inequality, we have

$$\left| \langle \pi_1 \tilde{\mu}_1(B) - \pi_2 \tilde{\mu}_2(B), \, \pi_1 \xi_{1,S} - \pi_2 \xi_{2,S} \rangle \right| \leq \|\pi_1 \tilde{\mu}_1(B) - \pi_2 \tilde{\mu}_2(B)\| \, \|\pi_1 \xi_{1,S} - \pi_2 \xi_{2,S}\|.$$

Bound the second factor:

$$\|\pi_1 \xi_{1,S} - \pi_2 \xi_{2,S}\| \leq \|\pi_1 \xi_{1,S}\| + \|\pi_2 \xi_{2,S}\| = \eta_1 \|\pi_1 \tilde{\mu}_1(B)\| + \eta_2 \|\pi_2 \tilde{\mu}_2(B)\|.$$

Therefore, the magnitude of the cross-term is bounded by

$$2\left| \langle \pi_1 \tilde{\mu}_1(B) - \pi_2 \tilde{\mu}_2(B), \, \pi_1 \xi_{1,S} - \pi_2 \xi_{2,S} \rangle \right| \leq 2\|\pi_1 \tilde{\mu}_1(B) - \pi_2 \tilde{\mu}_2(B)\| \left( \eta_1 \|\pi_1 \tilde{\mu}_1(B)\| + \eta_2 \|\pi_2 \tilde{\mu}_2(B)\| \right)$$

Putting everything together, we obtain

$$\left| 2\langle \pi_1 \tilde{\mu}_1(B) - \pi_2 \tilde{\mu}_2(B), \, \pi_1 \xi_{1,S} - \pi_2 \xi_{2,S} \rangle \right| \in \Theta \left( (\pi_1 + \pi_2)^2 \, \beta_t^A \right)$$

Since the cross-product is signed, it could contribute negatively to the mean difference. However, by the same argument it cannot exceed the leading term in magnitude. Unless the two terms perfectly cancel each other, the scaling with $t$ is dominated by the positive norms:

$$\|\mu_{1,S}(t) - \mu_{2,S}(t)\|^2 \in \Theta((\pi_1 + \pi_2)^2 \, \beta_t^A),$$

Putting everything together we obtain the asymptotic behaviour of the SNR lower bound:

$$SNR(c_1, c_2) \in \Theta \left( \frac{(\pi_1 + \pi_2)^2 \, \beta_t^A + (1 - (\pi_1 + \pi_2))^2 \, v^{2(t-t_0)}}{(\pi_1 + \pi_2)^2 \, \delta_t + (1 - (\pi_1 + \pi_2))^2 \, \left( \beta_t^A + v^{2(t-t_0)} \right)} \right)$$

To write it more clearly, define by $r^2 = \frac{(\pi_1 + \pi_2)^2}{(1 - (\pi_1 + \pi_2))^2}$:

$$SNR(c_1, c_2) \in \Theta \left( \frac{r^2 \, \beta_t^A + v^{2(t-t_0)}}{r^2 \, \delta_t + \left( \beta_t^A + v^{2(t-t_0)} \right)} \right)$$

For $r \to 0$ we recover the asymptotic behaviour of OOD data. For $r > 0$, the SNR *is guaranteed not to vanish in the TPT*. $\qquad \square$

**Corollary 4** (Asymptotic SNR with replay). *Under the conditions of Theorem 7, let* $r^2 = \frac{(\pi_1 + \pi_2)^2}{(1 - (\pi_1 + \pi_2))^2}$ *denote the buffer-weighted ratio of signal to residual OOD contribution. Then:*

- *In the limit* $r \to 0$ *(corresponding to no replay), the SNR asymptotically reduces to the OOD case, and old-task features remain vulnerable to drift in* $S^\perp$.

- *For any* $r > 0$ *(non-zero buffer fraction), the SNR is guaranteed not to vanish in the TPT as long as task-balanced replay is used. In particular, with increasing* $\beta_t$ *or weight decay, the limiting SNR satisfies*
$$\lim_{t \to \infty} SNR(c_1, c_2) \in \Theta(r^2),$$
*ensuring that replay effectively preserves linear separability between old-task classes in the NC subspace.*

