# OpenReview forum: "Heads collapse, features stay: Why Replay needs big buffers"
_ICLR.cc/2026/Conference — ICLR 2026 Poster_

### Official Review · Reviewer_hg1i · 2025-10-31

**Soundness:** 3
**Presentation:** 3
**Contribution:** 3
**Rating:** 6
**Confidence:** 2

**Summary:**

The author developed an asymptotic framework to analyze feature geometry with and without replay buffers. They demonstrate that replay can reliably alleviate deep forgetting (loss of feature separability), but does not alleviate shallow forgetting (misalignment between classifier weights and features). This work extends NC theory to multi head CL settings, characterizes the effects of buffer size and weight decay, and establishes a theoretical connection between CL and OOD. The empirical results of CIFAR100, Tiny ImageNet, and CUB-200 validate the theoretical findings.

**Strengths:**

Extends Neural Collapse analysis to continual learning and multi-head architectures—an unexplored direction. Uses asymptotic analysis and connects NC with OOD theory in a rigorous manner. Establishes a bridge between NC, CL, and OOD detection, enriching all three research domains.

**Weaknesses:**

While conceptually strong, it provides limited actionable guidance for improving CL performance.

**Questions:**

Why do we focus on discussing Multi Head Models? Is this model commonly used in modern continuous learning and multi task learning? Do you analyze whether the purpose of this model is to increase workload or has practical significance.

---

> ### Author Response · Authors · 2025-11-21
>
> We thank the reviewer for the feedback and questions. We believe we have addressed all the points raised by the reviewer with our new revision. We will now reply to the specific questions for completeness, but we refer the reviewer to the General Response for an overview.
>
> - *“While conceptually strong, it provides limited actionable guidance for improving CL performance.”*
> By uncovering the mechanisms behind the gap between deep and shallow forgetting, our paper lays the groundwork for algorithms that can effectively close this gap, potentially achieving high efficiency with small replay buffers. However, we believe that bridging this gap is not a matter of “simple fixes” (otherwise we would have implemented them ourselves); rather, it requires fundamentally rethinking the optimal representational structure for continual learning. We have included these reflections in the final remarks.
>
> - *“Why do we focus on discussing Multi-Head Models? Is this model commonly used...? Do you analyze whether the purpose of this model is to increase workload or has practical significance.”*
> Multi-head models are extremely common in Continual Learning and remain the standard choice whenever task labels are available. We refer the reviewer to standard reviews of the field [a] and to the popular “three scenarios” characterization of Continual Learning [b].
>
>
> [a] De Lange, Matthias, Rahaf Aljundi, Marc Masana, Sarah Parisot, Xu Jia, Aleš Leonardis, Gregory Slabaugh, and Tinne Tuytelaars. "A continual learning survey: Defying forgetting in classification tasks."
>
> [b] Gido M van de Ven, Tinne Tuytelaars, and Andreas S Tolias. Three types of incremental learning.

---

### Official Review · Reviewer_wGRs · 2025-10-31

**Soundness:** 3
**Presentation:** 3
**Contribution:** 3
**Rating:** 8
**Confidence:** 4

**Summary:**

This paper presents an asymptotic analysis of replay-based continual learning under the neural collapse phenomenon. The authors study how replay buffers affect shallow and deep forgetting. Empirically, they find that replay-based continual learning effectively mitigates deep forgetting but still suffers from shallow forgetting even when the replay buffers are large. They then use Neural Collapse theory to analyze the limiting geometry of features and heads in three continual learning setups. They also identify a connection between continual learning and OOD detection, showing that under weight decay, the distribution of OOD inputs converges to a degenerate null distribution. They also show the effect of replay in their framework, demonstrating that deep forgetting is not mitigated by the model with small replay buffer because of the approximation error when using the buffer distribution to approximate the true class distribution.

**Strengths:**

1. This work proposes a novel framework for replay-based continual learning. The results and implications are meaningful and helpful to the community.
2. The authors provide a sufficient and comprehensive theoretical analysis for replay-based continual learning, considering three different setups and showing the effect of replay.
3. The empirical study is consistent with the theoretical findings, across both real-world and simulated datasets.
4. The work is well structured and easy to follow for the readers.

**Weaknesses:**

1. In Theorems 1, 2, and 3, it seems that $\nu = 1 - \eta \lambda$ is required to be non-negative or greater than $-1$, but I do not find any explicit condition on $\nu$.

2. The explanation of why replay cannot strongly mitigate shallow forgetting is not convincing to me. The authors argue that the approximation error is the key reason, but there is no formal result to support this claim, which limits the contribution of this paper.

3. In the experimental results, the authors only vary the buffer size from $0\\%$ to $10\\%$. This seems insufficient to support their theoretical findings.

4. There are some typos and inconsistencies:
   1. Lines 139, 141. Two citations are missing.
   2. Line 315, "Theorem 6" should be "Theorem 2."
   3. Line 987, "class-il", "domain-il", and "task-il" should be written as "CIL," "DIL," and "TIL," consistent with other figures. Moreover, the task indices in Figure 12 should be positive integers.

5. The term “balanced replay” is unclear. I think “balanced replay” refers to the replay buffer being sampled in a balanced manner from the training set, rather than the buffer size being equal to the size of the training data.

**Questions:**

1. How to understand the theoretical results when $\eta \lambda \geq 2$?
2. Why are experiments conducted only for buffer sizes varying from $0\\%$ to $10\\%$?
3. What is the precise meaning of "balanced replay"?

---

> ### Author Response · Authors · 2025-11-21
>
> We thank the reviewer for the in-depth feedback and the great questions and suggestions, which allowed us to significantly improve our work.
>
> We believe we have addressed all the points raised by the reviewer in our revised manuscript. Below, we reply to your specific questions for completeness, but we also refer you to the General Response for a broader overview of the major updates.
>
> - “*How to understand the theoretical results when $\eta\lambda\ge 2$?*”
>     This is a good catch. We have carefully checked the theory to pinpoint exactly where this factor originates. In the proof of **Lemma 4**, which is central to our claims, we employ a first-order approximation of the feature evolution dynamics.
> This approximation clearly fails if the learning rate is too large (specifically if $\eta\lambda\ge 2$). We have therefore corrected our statements in the revised manuscript to explicitly require that the learning rate be “sufficiently small.” In standard practice, the learning rate is usually <1 and weight decay is well below 1 (often close to 0), so this condition is naturally satisfied in most realistic scenarios.
>
>
> - "*“The explanation of why replay cannot strongly mitigate shallow forgetting is not convincing to me... there is no formal result to support this claim.”* We really thank the reviewer for this comment. We actually shared your concern regarding the original explanation. Since the submission, we have worked hard to deepen our understanding of the mechanisms behind shallow forgetting.
> We believe we have finally found the right perspective. We now attribute this phenomenon to the **rank deficiency** of the observed (buffer) covariance matrix. Because the buffer data collapses so strongly, the covariance becomes rank-deficient, rendering the optimal linear decision boundary **under-determined**. This leads to "buffer-optimal" boundaries that do not generalize to the population. We found this concept so central that we have included this intuition directly in Figure 1. We kindly invite you to have another look at the revised paper, especially the detailed discussion of the Deep–Shallow forgetting gap in Section 4.3.
>
>
> - *“Why are experiments conducted only for buffer sizes varying from 0 to 10?”*
> Thank you for this question. To address this, we have now included a data point for 100% buffer size (the entire training set) in all relevant plots. We think our conclusions are even stronger now: showing the final convergence point further validates our theoretical model.
> The rationale for originally focusing on the 0–10% interval is that this is the "critical regime" for practical applications. In real-world scenarios, buffers beyond 10% are rarely used as performance tends to saturate; therefore, capturing the dynamics in this constrained setting is most relevant for designing memory-efficient systems.

---

### Official Review · Reviewer_5K6E · 2025-10-31

**Soundness:** 3
**Presentation:** 4
**Contribution:** 3
**Rating:** 8
**Confidence:** 2

**Summary:**

The paper analyzes how the replay buffer in continual learning scenarios influences model forgetting, distinguishing between shallow and deep forgetting. It further investigates these phenomena within the Neural Collapse framework, examining the geometric structure of the feature space and supporting the analysis with empirical results.

**Strengths:**

The paper is well written and clearly organized, with a pleasant and coherent flow of discussion. The topic addressed is novel and engaging. Moreover, the theoretical analysis is insightful and clearly explained.

**Weaknesses:**

While the theoretical discussion is sound and convincing, I have some concerns regarding the empirical analysis. First, it is unclear why the authors chose ResNet and ViT as reference models. It seems that the selected architectures could significantly influence the observed behaviors and results. If this is the case, the authors should explicitly discuss this aspect. Otherwise, a justification of why the chosen architectures do not affect the outcomes should be provided.  Along the same lines, the rationale behind considering both pretrained and from-scratch models is not entirely clear. In the case of pretrained models, it would be important to explain how the initialization was adapted to the continual learning setting, as mentioned in Section 1.1. Additionally, the discussion in Section 3.3.2 highlights the effect of weight decay, but the influence of other hyperparameters and architectural choices remains unexplored. Given their potential impact, especially in the context of deep forgetting, this omission seems non-negligible. The authors should include a discussion addressing this point to provide a more comprehensive understanding of the empirical results. Another aspect that would benefit from clarification is the adoption of the Neural Collapse (NC) framework. The authors should briefly discuss possible alternative frameworks and justify the choice of NC in this context.
Finally, the discussion on the distinction between multi-head and single-head settings could be improved by adding a short introductory explanation earlier in the paper to help readers unfamiliar with these concepts. In addition, Section 3 contains some citation issues, where “?” symbols appear instead of proper references, and these should be corrected.

**Questions:**

- Could the authors clarify the rationale behind choosing ResNet and ViT as reference architectures, and discuss how this choice might influence the observed behaviors and results?

- How were pretrained models adapted to the continual learning setting, and what motivated the comparison between pretrained and from-scratch training approaches?

- Beyond weight decay, have the authors examined the influence of other hyperparameters or architectural choices on the empirical results, particularly in relation to deep forgetting?

- What motivated the adoption of the Neural Collapse framework, and could the authors discuss potential alternative frameworks or justify why NC is particularly suitable for this analysis?

---

> ### Author Response · Authors · 2025-11-21
>
> We thank the reviewer for the in-depth feedback and the great questions and suggestions, which allowed us to significantly improve our work.
>
> We believe we have addressed all the points raised by the reviewer in our revised manuscript. Below, we reply to your specific questions for completeness, but we also refer you to the General Response for a broader overview of the major updates.
>
> - *"Could the authors clarify the rationale behind choosing ResNet and ViT as reference architectures, and discuss how this choice might influence the observed behaviors and results?”* Neural Collapse has been theoretically and empirically proven to emerge across a very wide range of architectures. Therefore, we do not believe our specific choice of architectures has significantly biased the observations, as our findings are fundamentally rooted in the universality of Neural Collapse.
> Our selection was driven by the need for high performance and broad relevance: ResNets are standard, high-performing baselines for these visual benchmarks, while ViTs were included to ensure our findings extend to modern attention-based architectures. In Figure 10 of the Appendix, we report the average accuracy of our models across buffer sizes; we invite the reviewer to verify that our scores are competitive with the state of the art in Continual Learning [a].
> - *“How were pretrained models adapted to the continual learning setting, and what motivated the comparison between pretrained and from-scratch training approaches?”*
> Leveraging pre-trained models has become standard practice in modern Continual Learning. To adapt these models to our specific settings, we replace the original pre-training head with a new, task-appropriate classification head (or multi-head structure) and fine-tune the feature extractor on the incoming data stream.
> We have clarified our motivation in the revised Section 2: given that prior work demonstrates the remarkable robustness of pre-trained representations against both deep and shallow forgetting [b,c], it was natural to evaluate how this intrinsic stability interacts with replay mechanisms. Furthermore, we have added a new ablation in the Appendix (Section A.3.1) showing that pre-trained models converge significantly faster to Neural Collapse structures compared to models trained from scratch—a novel insight that helps explain their geometric behavior.
> - *“Beyond weight decay, have the authors examined the influence of other hyperparameters or architectural choices on the empirical results, particularly in relation to deep forgetting?”*
> This question prompted us to revisit the Neural Collapse literature to align our experimental scope with established practices. Most of the literature experiments only with weight decay, the dataset size and the model architecture itself. Given that we had already evaluated the effect of weight decay (Figure 5,20,21) and buffer size (Figure 15), we decided to expand our study of the effect of the architecture. In our main experiments, the feature dimension $d$ was significantly larger than the number of classes $K$ ($d \gg K$), which allows for the standard Simplex ETF geometry. However, in many practical applications (e.g., language modeling), the reverse is true ($d < K$). To address this, *we have added a new ablation study* on models where the feature space is strictly lower-dimensional than the output space. We have summarized our findings in the new Section A.3.2 of the Appendix.
> - *“What motivated the adoption of the Neural Collapse framework, and could the authors discuss potential alternative frameworks or justify why NC is particularly suitable for this analysis?”*
> We thank the reviewer for this sharp question. It highlighted an implicit assumption in our work that we had not fully articulated in the initial submission.
> We have now added a dedicated paragraph in *Section 3* to explicitly justify this choice. To summarize our rationale:
> standard evaluation in Continual Learning measures performance strictly at the completion of each task (a series of discrete snapshots). Therefore, while the transient *dynamics* of training are important, the magnitude of forgetting is ultimately defined by the network's *asymptotic configuration*. Neural Collapse is uniquely suited for this analysis because it is the only framework that rigorously characterizes the precise geometric structure to which features converge at the exact moments where forgetting is measured.
>
> [a] Buzzega, Pietro, Matteo Boschini, Angelo Porrello, Davide Abati, and Simone Calderara. "Dark experience for general continual learning: a strong, simple baseline."
>
> [b] Ramasesh, Vinay Venkatesh, Aitor Lewkowycz, and Ethan Dyer. "Effect of scale on catastrophic forgetting in neural networks."
>
> [c] Mehta, Sanket Vaibhav, Darshan Patil, Sarath Chandar, and Emma Strubell. "An empirical investigation of the role of pre-training in lifelong learning."

---

### Author Response · Authors · 2025-11-21
**General Response to All Reviewers**

We thank the reviewers for their constructive feedback. We have revised the manuscript to address the specific points raised in the reviews, alongside additional refinements developed since the initial submission. We believe these revisions have substantially improved the paper, and we invite you to examine the updated text. To assist in this process, we have outlined the major changes below, which should also respond to the main questions raised in the reviews:


* **Formalized Mechanism for the Deep–Shallow Gap:** We have updated the theoretical explanation for the persistence of shallow forgetting. The revised analysis grounds this phenomenon in the *rank deficiency* of the buffer covariance matrix, which renders the decision boundary *under-determined*. This mechanism is now illustrated in Figure 1, with explicit measurements of covariance rank deficiency provided in Figure 6.
* **Structural & Notational Refinements:** The manuscript structure has been reorganized for improved flow. Key changes include:
    * **Formalized Hypotheses:** The two central hypotheses underpinning the theoretical model are now explicitly isolated within the text.
    * **Expanded Preliminaries:** Background sections now include a formal definition of "replay," clear notation for NC quantities, and a distinction between *population statistics* (full data) and *observed statistics* (buffer data).
    * **Clarifications:** Added concise explanations for "balanced sampling" and "multi-head" architectures.
* **Full Replay Baseline:** All relevant plots now include a baseline corresponding to *100% buffer size* (full training set) to visualize the asymptotic convergence point of the replay curves.
* **Expanded Experimental Scope:** We have added several new analyses and ablations:
    * **Pre-training vs. Scratch:** (Section A.3.1) We analyze convergence to NC in pre-trained models, observing significantly faster collapse compared to models trained from scratch.
    * **Alternative Replay Algorithms:**  (Section A.4)  We demonstrate the evolution of deep–shallow forgetting curves under other prominent replay strategies.
    * **Class Mean Norms:**  (Section A.4)  now details the evolution of class mean norms as a function of weight decay and buffer size.
    * **Head Initialization:**  (Section A.3.3) We find that while head initialization affects the temporal growth of feature norms, it does not substantially alter forgetting outcomes.
    * **Low Feature-Dimension Regime ($d < K$):**  (Section A.3.2)  We investigated the scenario where the feature dimension is smaller than the number of classes. We observe that while the rigid Simplex ETF structure ($\mathcal{NC}2$) degrades in this regime—consistent with an expected geometric shift—the **deep–shallow forgetting gap persists**. This confirms the gap is not contingent on a specific ETF geometry.
* **Visual Enhancements:** All figures have been updated for better readability, and theoretical results are now highlighted for easier navigation.

We remain fully available during the discussion period and are eager to engage with any further feedback. We welcome any additional ideas or suggestions for experiments that could further strengthen the paper and are happy to discuss them.

---

### Comment · Area_Chair_bvTd · 2025-11-22

Dear reviewers,
Please check the authors’ responses. As there are differing opinions about some aspect of the paper, it would be appreciated if you could evaluate—based on all comments—whether the authors have adequately addressed the main concerns.
Br,

---

### Meta-Review · Area_Chair_MLyf · 2026-01-06

**Summary:**

The paper studies continual learning with replay. The authors show that replay size has different effects on deep (feature-level) forgetting and shallow (classifier-level) forgetting. Using the neural collapse framework, they provide a theoretical analysis explaining why deep forgetting is mitigated while shallow forgetting persists. Experimental results on CIFAR-100, Tiny ImageNet, and CUB-200 support the theoretical findings.

**Reviewer Concerns:**

Overall, this paper receives positive comments. Most concerns relate to clarifying the experimental settings, the motivation for using the Neural Collapse framework, and the definitions of parameters and concepts in the theoretical analysis (e.g., balanced replay). These concerns are likely to be addressed in the rebuttal.

**Reviewer Scores:**

The initial reviewer scores before the rebuttal are 8, 8, and 6. Since most concerns are likely to be addressed in the rebuttal, the reviewers may at least maintain their scores. The reviewers report low confidence, so it is unclear whether they would be willing to further increase their scores.

---

### Decision · Program_Chairs · 2026-01-26

Accept (Poster)